



# A new discrete multiplicative random cascade model for downscaling intermittent rainfall fields

Marc Schleiss[1]

[1]Department of Geoscience & Remote Sensing, Delft University of Technology

**Correspondence:** Marc Schleiss (m.a.schleiss@tudelft.nl)

**Abstract.** Spatial downscaling of rainfall fields is a challenging mathematical problem for which many different types of methods have been proposed. One popular solution consists in redistributing rainfall amounts over smaller and smaller scales by means of a discrete multiplicative random cascade (DMRC). This works well for slowly varying, homogeneous rainfall fields but often fails in the presence of intermittency (i.e., large amounts of zero rainfall values). The most common workaround
in this case is to use two separate cascade models, one for the occurrence and another for the intensity. In this paper, a new and simpler approach based on the notion of equal-volume areas (EVAs) is proposed. Unlike classical cascades where rainfall amounts are redistributed over grid cells of equal size, the EVA cascade splits grid cells into areas of different sizes, each of them containing exactly half of the original amount of water. The relative areas of the sub-grid cells are determined by drawing random values from a logit-normal cascade generator model with scale and intensity dependent standard deviation. The process
ends when the amount of water in each sub-grid cell is smaller than a fixed bucket capacity, at which point the output of the cascade can be re-sampled over a regular Cartesian mesh. The present paper describes the implementation of the EVA cascade model and gives some first results for 100 selected events in the Netherlands. Performance is assessed by comparing the outputs of the EVA model to bilinear interpolation and to a classical DMRC model based on fixed grid cell sizes. Results show that on average, the EVA cascade outperforms the classical method, producing fields with more realistic distributions, small-scale
extremes and spatial structures. Improvements are mostly credited to the higher robustness of the EVA model to the presence of intermittency and to the lower variance of its generator. However, improvements are not systematic and both approaches have their advantages and weaknesses. For example, while the classical cascade tends to overestimate small-scale extremes and variability, the EVA model tends to produce fields that are slightly too smooth and blocky compared with observations.

## 1 Introduction

Stochastic rainfall downscaling algorithms are statistical methods designed to enhance the resolution of coarse-scale rainfall observations for use in hydrological modeling, weather prediction or flood risk analyses. Their simplicity and low computational cost mean that large ensembles of possible realizations for a single input field can be generated. This leads to a better representation of measurement errors and model uncertainties compared to physical downscaling as well as a more realistic representation of small-scale variability. However, the statistical nature of the approach means that one needs to find a good





balance between model complexity and performance (e.g., the realism of the distributions and spatial patterns that can be reproduced).

Popular statistical downscaling methods for global and regional climate models include various forms of transfer functions and quantile matching (Li et al., 2010; Teutschbein and Seibert, 2012; Langousis et al., 2016), machine learning (Jha et al.,
2015; He et al., 2016) and a multitude of hybrid physical-statistical and autoregressive models (e.g., Lisniak et al., 2013; Bechler et al., 2015; Xu et al., 2015). Another important family revolves around the notion of self-similarity, generalized scale-invariance and multiplicative random cascades (e.g., Lovejoy and Mandelbrot, 1985; Schertzer and Lovejoy, 1987; Gupta and Waymire, 1993; Menabde et al., 1997; Schertzer and Lovejoy, 2011). The main appeal of these techniques is that they require a very small number of model parameters, many of which can be inferred directly from the coarse-scale data. Also, the
framework itself is very flexible, applying to all kinds of rainfall inputs from time series to spatial and space-time fields (e.g., Deidda, 2000; Menabde and Sivapalan, 2000; Kang and Ramirez, 2010; Raut et al., 2018).

One long-standing and still unresolved issue of random multiplicative cascade models applied to rainfall concerns the question of how to properly deal with zero-rainfall values. Zeros are fundamentally incompatible with the notion of self-similarity and multiplicative random cascades (Gupta and Waymire, 1993). They must be artificially introduced into the cascade, for
example by setting a hard threshold on the minimum detectable intensity (e.g., Pathirana et al., 2003) or by modifying the cascade model in such a way that grid cells below a given intensity only have a finite, predetermined probability to survive at each cascade level (Gires et al., 2013). Another workaround consists in applying two separate cascade models for the occurrence and intensity (e.g., Over and Gupta, 1996; Olsson, 1998; Paulson and Baxter, 2007; Schmitt, 2014; Lombardo et al., 2017). However, this requires many additional model parameters to be estimated from the data, which can be challenging numerically
and increases the risks of overfitting. Regardless of how they are handled, zero rainfall values are likely to negatively impact the scaling properties of rainfall, making it difficult to retrieve reliable model parameters in the first place (Kedem and Chiu, 1987; Schmitt et al., 1998; Veneziano et al., 2006; de Montera et al., 2009; Gires et al., 2012; Veneziano and Lepore, 2012; Mascaro et al., 2013).

Given the numerous challenges mentioned above, there is a strong incentive to design new simple multiplicative cascade
models capable of handling rainfall fields with high levels of intermittency. Particular attention is given to parsimonious models with maximum 3 parameters whose values can be inferred directly from the coarse-scale data. One promising avenue explored in this paper revolves around the notion of "equal-volume areas" (EVAs), a natural extension of the inter-amount times concept introduced in the context of time series analysis by Schleiss and Smith (2016). The theoretical foundation for this work is motivated by recent studies by Schleiss (2017) and ten Veldhuis and Schleiss (2017) who showed that intermittent rainfall and
flow time series scale better when sampled adaptively rather than with a fixed frequency. The hope is that by switching to an adaptive sampling strategy, the mathematical challenges associated to the presence of zero rainfall values can be alleviated, thus leading to more robust cascades and more realistic rainfall fields after downscaling. The present study describes the implementation of this idea to the case of 2D rainfall fields and discusses its advantages and limitations with respect to traditional random cascades based on intensity.





The rest of this paper is structured as follows: Section 2 introduces the new EVA model, including the splitting rule, cascade generator and parameter estimation. In Section 3, the potential of the new cascade is demonstrated by applying it to radar rainfall snapshots collected over the Netherlands. First, the parameterization problem is discussed. Then, the performance is evaluated by means of controlled simulation experiments during which 100 high-resolution rainfall fields are aggregated to

5 coarser scales and subsequently downscaled back to their original resolution. Results are compared to two alternative downscaling techniques (i.e., bilinear interpolation and a classical random cascade based on intensity). The advantages and limitations of the model as well as possible extensions are discussed in Section 5 and the conclusions are given in Section 6.

## 2 Methods

### 2.1 A brief introduction to discrete multiplicative random cascades

Discrete multiplicative random cascades (DMRC) are statistical downscaling techniques designed to enhance the resolution of a coarse-scale rainfall field to a desired fine-scale target resolution. For spatial cascades, this is done by successively splitting the dimensions of coarse-scale grid cells by 2 (or 4 depending on type of cascade) according to a predefined branching rule. For example, one large $16\times16$ grid cell might be divided into 2 sub-grid cells of size $8\times16$ km$^2$ at the first level of the cascade which in turn, will be divided into 4 grid cells of size $8\times8$ km$^2$ at the next level. The splitting process is repeated iteratively

until the desired target resolution $l_x \times l_y$ is reached. During a split, each of the generated sub-grid cells receives a random fraction of the total rainfall amount in the parent grid. Redistribution takes place according to some multiplicative weights $W_1 \geq 0$ and $W_2 \geq 0$ drawn from a probability distribution $\Gamma$ called the cascade generator. In micro-canonical models, the sum of the weights associated to each split is forced to 1, thus ensuring that the total rainfall amount in each grid cell is preserved. By contrast, in canonical cascades, only the average rainfall intensity over a large number of grid cells needs to be preserved.

This has some advantages in terms of modeling but generally results in lower performance than micro-canonical cascades (e.g., Hingray and Ben Haha, 2005). For the sake of completeness, it should also be mentioned that other types of cascades have been proposed to downscale rainfall, such as those based on continuous in scale multifractal cascades (Lovejoy and Schertzer, 2010a, b). However, these are outside the scope of this paper which focuses on discrete micro-canonical random cascades.

As pointed out by Rupp et al. (2009), differences in performance between cascade models primarily relate to what proba-

25 bility distribution is chosen for $\Gamma$ and how rainfall amounts are re-assigned to sub-grid cells during the splits. In the simplest possible setup, the probability distribution of the generator remains the same across the entire cascade. However, rainfall fields downscaled with such an approach often exhibit unrealistically high small-scale variability and extremes. Consequently, many authors recommend to use cascade generators whose distribution depends on the spatial/temporal dimensions of the grid cells that are being split or on the average rainfall intensity within them (Rupp et al., 2009; Licznar et al., 2011). This is consistent

with our own findings which show that the variance of empirical cascade weights often depends on the spatial scale and the rainfall intensity.





## 2.2 Description of the EVA cascade model

Let $R_1, \ldots, R_N$ (in mmh$^{-1}$) denote a coarse scale rainfall intensity field over a regular Cartesian grid composed of individual grid cells of size $L_{x_i} \times L_{y_i}$ (in km$^2$), where $L_{x_i}$ and $L_{y_i}$ (in km) denote the horizontal and vertical dimensions respectively and $N$ is the total number of grid cells in the field. Let $A_i = L_{x_i} \cdot L_{y_i}$ denote the areas (in km$^2$) of the individual grid cells.

The relation between intensity $R_i$ (in mmh$^{-1}$), area $A_i$ (in km$^2$), volume $V_i$ (in millions of liters) and temporal aggregation time scale $\Delta t$ (in hours) is given by:

$$R_i = \frac{V_i}{A_i \cdot \Delta t} \tag{1}$$

In a classical cascade model, grid cells of area $A_i$ are divided in two sub-grid cells of equal areas $A_{(i,1)} = A_{(i,2)} = \frac{A_i}{2}$. The rainfall volumes $V_{(i,1)}$ and $V_{(i,2)}$ of the sub-grid cells are determined by multiplying $V_i$ by random weights $W_1 \geq 0$ and

$W_2 = 1 - W_1$ drawn from the cascade generator model $\Gamma$:

$$\text{Classical cascade :} \quad \begin{pmatrix} A_i \\ V_i \end{pmatrix} \rightarrow \left\{ \begin{pmatrix} A_{(i,1)} = \frac{A_i}{2} \\ V_{(i,1)} = W_1 \cdot V_i \end{pmatrix} ; \begin{pmatrix} A_{(i,2)} = \frac{A_i}{2} \\ V_{(i,2)} = W_2 \cdot V_i \end{pmatrix} \right\} \tag{2}$$

The random quantities in this case are the rainfall volumes $V_i$ (or equivalently the rainfall intensities) at each level and the area of the grid cells plays the role of the scale $\lambda$. This is the most natural choice for downscaling applications and will be referred to as the classical approach in this paper. The main drawback of the classical approach is that the conditional

probability distribution function of $V_i$ given $A_i > 0$ has a mixed distribution with atom at zero:

$$\mathbb{P}[V_i = 0 \mid A_i > 0] > 0 \tag{3}$$

where $\mathbb{P}$ denotes the probability. Moreover, the probability that $V_i$ equals zero knowing $A_i > 0$ increases as the area tends to zero. To reproduce such behavior, the classical cascade generator model $\Gamma$ must include a mechanism through which (some) of the weights can be set to zero during the splitting process (usually at the expense of additional model parameters). This is

far from trivial as one needs to make sure that the cascade does not remove all rainy areas during the downscaling and does not introduce zeros immediately next to grid cells with very high rainfall intensities (Olsson, 1998).

The main contribution of this paper is to show that many of the issues associated with zero rainfall values can be avoided by adopting a slightly different representation of rainfall based on the notion of equal-volume areas. In the EVA framework, the scale $\lambda$ is given by the total rainfall volume $V_i = A_i \cdot R_i \cdot \Delta t > 0$ contained in a grid cell and the random quantities that are

being downscaled are the areas $A_i$ needed to accumulate fixed volumes of water. At each split, the total volume of water $V_i$ in a grid cell is divided by two and equally redistributed over two sub-grid cells of different areas. The areas $A_{(i,1)}$ and $A_{(i,2)}$ of the two sub-grid cells are determined by drawing random weights $W_1$ and $W_2 = 1 - W_1$ from a cascade generator $\Gamma_{\text{EVA}}$ with predetermined probability distribution. A small diagram illustrating this process is provided in Figure 1.

$$\text{EVA cascade :} \quad \begin{pmatrix} A_i \\ V_i \end{pmatrix} \rightarrow \left\{ \begin{pmatrix} A_{(i,1)} = W_1 \cdot A_i \\ V_{(i,1)} = \frac{V_i}{2} \end{pmatrix} ; \begin{pmatrix} A_{(i,2)} = W_2 \cdot A_i \\ V_{(i,2)} = \frac{V_i}{2} \end{pmatrix} \right\} \tag{4}$$



Note that by convention, splits always occur perpendicular to the longest grid cell dimension, that is, splitting horizontally if $L_x \leq L_y$ and vertically otherwise. Splitting is applied iteratively, until the total rainfall volume in a grid cell is lower than a fixed bucket capacity $\varepsilon_V > 0$ which denotes the smallest rainfall volume that can be detected at the target resolution. The latter can be prescribed by end-user requirements or imposed to match known instrumental limitations such as the capacity

of a tipping bucket rain gauge or the sensitivity of a weather radar. The smaller the bucket capacity, the larger the number of cascade levels and sub-grid cells. Note that the rule above also applies to grid cells for which $V_i = 0$, which by definition do not need to be split and are kept "as is" until the end of the cascade.

The main advantage of the EVA approach is that the areas needed to accumulate a positive rainfall volume $V_i > 0$ can never be zero. This may sound obvious at first but can make a substantial difference when dealing with highly intermittent rainfall

fields:

$$\mathbb{P}[A_i = 0 \mid V_i > 0] = 0 \qquad (5)$$

Finally, note that by construction, the EVA cascade described above implements an adaptive spatial sampling of the coarse-scale rainfall field, very similar to that of a quadtree (Shankar and Hutchinson, 1990). The cascade decomposes a regular two-dimensional rainfall field into grid cells of variable sizes, with fewer and larger grid cells in areas of low rainfall intensity

and more numerous and smaller grid cells in areas of large rainfall intensities. The redistribution rule ensures each of the generated sub-grid cells contains a strictly positive rainfall amount, no matter its size or at which level of the cascade it was produced. Zeros are not coded explicitly into the field, making it unnecessary to model their distribution and structure. The downside of the approach is that the output of the cascade consists of grid cells of variables sizes. From a practical point of view, it may therefore be necessary to re-sample the output of the EVA cascade onto a regular Cartesian mesh with fixed spatial

resolution, at which point the zero rainfall values will become apparent. This process, also known as "regridding", is commonly encountered in geophysical image mapping and various computationally efficient solutions have been proposed for it. Here, we consider the simple case of regridding an irregular rectilinear grid to a regular Cartesian mesh composed of square pixels of size $l_x \times l_y$ centered around $(x_i, y_i)$. The total rainfall amount $V(x_i, y_i)$ in a target pixel centered around $(x_i, y_i)$ is given by the sum of all rainfall amounts in the irregular source field times the ratio of overlapping areas with the target pixel:

$$V(x_i, y_i) = \sum_{j=1}^{m} \lambda_{ij} V_i \qquad (6)$$

where $0 \leq \lambda_{ij} \leq 1$ denotes the fractional area-overlap of the target grid cell $i$ with the source cell $j$ and $m \in \mathbb{N}$ is the total number of grid cells generated by the cascade. Fractional overlaps for rectangular grid cells are easy to calculate, making this step very efficient. In the end, all re-sampled rainfall amounts $V(x_i, y_i)$ below the minimum detectable threshold $\varepsilon_V > 0$ are set to zero, similarly to how they would appear in real measurements. Note that this threshold is not imposed on the cascade

output itself (which does not contain any zeros) but only on the re-sampled quantities. Because of this, the frequency of zero rainfall values and their location in the domain will depend on the spatial scale at which the field is displayed. The latter can be changed at any time depending on user requirements without having to run another random cascade. In fact, an irregular grid





combined with a final re-sampling step for visualization constitutes a very natural way of modeling a scale-dependent process like rainfall.

## 2.3  Splitting Rule

The way grid cells are split at each level plays a crucial role in determining the spatial structure of the downscaled field.

Independently of the used cascade generator, for any weight $0 < W < 1$, there are only two possibilities to split a grid cell. In the case of vertical splits, the left sub-grid cell can be assigned the area $W \cdot A_i$ (corresponding to an intensity of $\dfrac{R_i}{2W \cdot A_i}$ or, conversely, the complementary value $(1 - W) \cdot A_i$. The splitting rule is a set of instructions for determining which side gets assigned the lowest area or, equivalently, the highest rainfall intensity. To preserve the overall spatial structure and coherency of the rainfall field during downscaling, knowledge about the rainfall intensity in surrounding grid cells is required. This is

achieved by performing inverse distance interpolation of the coarse-scale rainfall intensity field on the left/right (for horizontal splits) or top/bottom (for vertical splits) sides of each grid cell. At each split, the side with the highest interpolated rainfall value gets assigned the largest intensity (i.e., the smallest area). Since the spatial distribution of the rainfall intensity over the domain changes after each split, the interpolated values need to be updated regularly to take into account the newly generated fine-scale rainfall patterns. Without this regular update, downscaled fields would rapidly lose their spatial coherency. Ideally, a

new interpolation should be performed after each split, especially at the beginning of the cascade when grid cells are still large. To save time at later stages, it is also acceptable in practice to update the interpolation only once in a while, for example after a fixed number of splits or at the end of each new cascade level. Results show that this strategy can save precious time when the number of sub-grid cells becomes large while only marginally affecting the small-scale structure of the downscaled fields.

Note that the splitting rule as defined above only takes into account the rainfall values in surrounding grid cells without

influencing the cascade weights themselves. Its only purpose is to ensure that, as we go through the cascade, water gets redistributed in a way that is spatially coherent with respect to the coarse-scale observations and all previously generated grid cells during the cascade. This is particularly important in the first stages of the cascade, when rainfall amounts can get redistributed very unevenly. The choice of the interpolation scheme is not critical as long as it provides a relatively smooth estimate of the rainfall distribution over the domain. To limit the computational cost associated with interpolation, only a

limited number of nearest neighbors can be used.

## 2.4  The cascade generator

The probability distribution of the cascade generator is a crucial component of any discrete multiplicative random cascade (Over and Gupta, 1994; Ossiander and Waymire, 2000). Without any explicit physical law governing the redistribution of precipitation over scales, choosing an appropriate generator model can be a rather subjective task. Consequently, a wide range of possible

distributions have been proposed so far, from uniform (Olsson, 1998) to log-normal, Beta (Ahrens, 2003; Molnar and Burlando, 2005; Paulson and Baxter, 2007) and log-Levy (Schertzer and Lovejoy, 2011). Beyond the ability of the generator to reproduce observed scale-invariance in data, other important factors to consider are simplicity and ease of interpretation. One distribution

(c) Author(s) 2019. CC BY 4.0 License.





that satisfies all these criteria and will be used in this study is the logit-normal distribution:

$$\ln\left(\frac{W}{1-W}\right) \sim N(\mu, \sigma^2) \tag{7}$$

where $\mu \in \mathbb{R}$ and $\sigma \geq 0$ represent the mean and standard deviation of an underlying Gaussian random variable. Further simplifications can be made by assuming that the cascade weights are symmetrically distributed around 0.5, which forces $\mu$ to be

zero.

The logit-normal generator model may not necessarily be optimal for all types of events and across all scales. But the results obtained within this study show that it produces a fair enough approximation of empirical cascade weight distributions to be useful in practice. The main practical advantages are that the distribution is continuous, supported over the open unit interval (0,1) and that it can be easily simulated through its analytical link with the Gaussian distribution. The most important

advantage of all, however, lies in the ease of interpretation of the model parameter $\sigma$, which measures the spread of the underlying Gaussian and directly relates to the sub-grid variability (i.e., the intermittency) of the rainfall process within a given grid cell. Figure 2 illustrates this point by showing the density function of a logit-normal cascade generator $W$ with $\mu = 0$ for 4 different values of $\sigma$. We see that for small values of $\sigma$, the distribution tends to a delta function centered around 0.5. This translates to low spatial variability of the rainfall rate within a grid cell and results in grid cells splitting up very evenly. On the

other hand, as $\sigma \to \infty$, the density of $W$ progressively moves away from 0.5, tending to 0 almost everywhere except for two small symmetric intervals near 0 and 1 (without ever reaching these limits). This corresponds to high spatial variability and high intermittency and means that grid cells split up very unevenly.

Since $\mu = 0$ is fixed, the only parameter needed to define the full distribution of the cascade generator is $\sigma$. Previous research on discrete multiplicative random cascades has shown that the empirical distribution of $W$ usually depends both on the intensity

and spatial scale (e.g., Over and Gupta, 1994; Olsson, 1998; Marani, 2005; Rupp et al., 2009; De Luca, 2014). The analyses conducted within this study confirm these previous findings, showing that within the EVA framework, on average, the spread of the cascade weights increases with area $A$ and decreases with intensity $R = \frac{V}{A \cdot \Delta t}$. Based on these empirical observations, a simple power-law model for expressing the standard deviation $\sigma$ of the cascade generator $W$ is proposed:

$$\sigma(A, R) = a \cdot R^{-b} \cdot A^c \tag{8}$$

where $A$ (in km$^2$) denotes the area of the grid cell to be split, $R$ (in mmh$^{-1}$) is the intensity (for a given area $A$ and temporal resolution $\Delta t$) and $a > 0, b > 0, 0 < c < b$ are three model coefficients. Note that because the amount of water is halved at each split, according to Equation (8), the fate of individual grid cells in the cascade will be determined by how quickly their area decreases with respect to their intensity. In fact, if we impose $b > c$ and let the cascade run for a long enough time, only two possible outcomes can result: either $\sigma \to 0$ or $\sigma \to \infty$. In the first case, grid cells of area $A_i$ end up splitting in two almost equal

areas $A_{(i,1)} \approx A_{(i,2)} \approx \frac{A_i}{2}$. The cascade generator for the generated sub-grid cells after the split will have standard deviation:

$$\sigma(A_{(i,1)}, R_{(i,1)}) = \sigma(A_{(i,2)}, R_{(i,2)}) = 2^{-c} \cdot \sigma(A_i, R_i) < \sigma(A_i, R_i) \tag{9}$$

Therefore, grid cells at subsequent cascade levels will split more and more evenly, converging to a fixed rainfall intensity. In the second case (i.e., $\sigma \to \infty$), grid cells split up very unevenly. Without loss of generality, we can assume that $A_{(i,1)} \approx A_i$ and





$A_{(i,2)} \approx 0$. The standard deviations of the cascade generators for the first sub-grid cell is then given by:

$$\sigma(A_{(i,1)}, R_{(i,1)}) \approx \sigma(A_i, \frac{R_i}{2}) = 2^b \cdot \sigma(A_i, R_i) > \sigma(A_i, R_i) \tag{10}$$

while the one of the second sub-grid cell will be $\sigma(A_{(i,2)}, R_{(i,2)}) \approx 0$. The first sub-grid cell will therefore continue to split up very unevenly as we progress through the cascade while the second sub-grid cell will have a higher intensity and therefore

split up into two almost equal parts at subsequent levels. The final result of this process is a bounded cascade in which some grid cells have areas converging to a strictly positive lower bound (defining the dry areas) while all others converge to strictly positive rainfall rates.

## 2.5 Sample estimation of cascade generator model

An important advantage of micro-canonical model is that the distribution of the cascade weights can be studied directly from

the data through the calculation of empirical breakdown coefficients (Cârsteanu and Foufoula-Georgiou, 2016; Licznar et al., 2015). The latter are estimated by successively aggregating grid cells in the input field to larger spatial scales and by studying how the rainfall volumes in aggregated grid cells split up as a function of area and rainfall intensity. For example, an input field of $1 \times 1$ km$^2$ resolution might be aggregated to blocks of size $1 \times 2$ km$^2$, $2 \times 1$ km$^2$, $2 \times 2$ km$^2$, $4 \times 2$ km$^2$ etc..., each of which can be split in two equal sub-areas to analyze the redistribution of rainfall volumes inside them. For the EVA framework, the

procedure is similar except that we are interested in determining the sub-area needed to accumulate half of the rainfall amount in the parent grid cell. The main drawback compared with the classical approach is that due to the fixed grid spacing, the sub-areas can not be determined exactly but must be approximated by linear interpolation, similarly to the procedure described in Equation (4) of Schleiss (2017). For example, a grid cell of size $16 \times 8$ km$^2$ might split in two sub-grid cells of size $7.3 \times 8$ km$^2$ and $8.7 \times 8$ km$^2$ where the first dimension 7.3 km was determined by interpolating the rainfall volumes contained in the smaller

grid cell of size $7 \times 8$ km$^2$ (containing slightly less than half the amount) and the one immediately above of size $8 \times 8$ km$^2$ (which contains more than half). The additional interpolation step means that the empirical breakdown coefficients of small grid cells will be affected by larger sampling uncertainties compared with large grid cells

In the classical cascade model, no linear interpolation is needed. However, some of the rainfall volumes in the sub-grid cells may be zero (i.e., one size receives all the rain). Such splits are fundamentally incompatible with the logit-normal model

prescribed in Equation (7). To avoid numerical issues when evaluating $\ln(W)$, one can set the weights to a small positive value close to zero or simply ignore the problematic splits (which is the approach adopted in this paper). Because some splits are ignored during parameter estimation, the cascade generator model for the classical cascade model and highly intermittent rainfall fields is likely to be biased.

Once the empirical breakdown coefficients have been determined from the sample, the last step consists in estimating the

three model parameters $a$, $b$ and $c$ in Equation (8). To do this, the empirical breakdown coefficients are grouped in classes according to the total area $A$ and rainfall intensity $R$ of the parent grid cell that generated them. For the area $A$, the spacing between classes is imposed by the spatial resolution of the input field. For the intensity, the number of classes that can be formed depends on how many empirical breakdown coefficients are available at a given spatial scale. In our case, 30 regularly





spaced intensity classes were used for each value of $A$. Moreover, each class of $(A,R)$ needed to contain at least 50 empirical breakdown coefficients in order to estimate the standard deviation $\sigma(A,R)$ of the underlying logit-normal distribution. In the end, once $\sigma(A,R)$ has been estimated for all values of $A$ and $R$, the coefficients $a$, $b$ and $c$ of the power-law model in Equation (8) were estimated through non-linear least square fitting (implemented in the function nls() in R).

## 2.6  Benchmarks

While the EDA downscaling technique is the main focus of this paper, two additional spatial downscaling techniques were considered for comparison purposes. The first is bilinear interpolation, implemented in the function "interp.surface()" of the R package "fields" (Douglas Nychka et al., 2017). Bilinear interpolation is a deterministic non-parametric downscaling method. It makes no assumption about the structure and distribution of the data, making it very robust. However, because it is an interpolation technique, it tends to generate fields that are too smooth compared with the observations. Its main purpose here is to serve as an easy to beat baseline against which the added-value of the random cascade models can be assessed.

The second benchmark is a classical micro-canonical discrete multiplicative random cascade based on rainfall intensity as described in Equation (2). To ensure fair comparisons, the classical cascade model is set up to be a perfect replicate of the new EVA model. It uses the same logit-generator model, the same splitting rules and the same power law model as in Equation (8), albeit with different $a$, $b$, and $c$ coefficients. Note that the classical cascade is run without performing any separation between the occurrence and intensity process. Dry and rainy regions are delineated at the end by imposing a fixed threshold on the minimum detectable rainfall volume at the target resolution, similarly to what is done in the EVA cascade. This may not be state-of-the-art but ensures a fair comparisons and makes it easier to outline the strengths and limitations of both approaches.

To assess performance, synthetic experiments on high-resolution radar rainfall fields were performed. During these experiments, 100 different 5-min radar rainfall snapshots from the operational Dutch national C-band radar composite over an area of $128 \times 128$ km$^2$ near Rotterdam were aggregated from their original spatial resolution of $1 \times 1$ km$^2$ to square blocks of size $2 \times 2$ km$^2$, $4 \times 4$ km$^2$ and $8 \times 8$ km$^2$ (see Figure 3 for events 1-4). Then, the fields were downscaled back to their initial resolution of $1 \times 1$ km$^2$. For each event, 100 different realizations of the random cascades were generated to get an estimate of the ensemble spread. The threshold used to distinguish dry from rainy grid cells at the target resolution was set to 0.1 mmh$^{-1}$ (corresponding to a bucket capacity of $\varepsilon_V = 8333$ liters for each grid cell of size 1 km$\times$1 km$\times$5 min), to match the minimum measurable rainfall intensity in the Dutch radar product. Performance is assessed both visually and quantitatively using a set of standard statistical metrics (e.g., bias, root-mean square error, quantiles, coefficient of determination and variograms). Among the 100 radar snapshots used for performance evaluation, the first four were selected for in-depth analyses (see Figure 3 and Table 1 for more details). Two of them (i.e., events 2 and 4) are characterized by widespread, predominantly stratiform and homogeneous rain with low rainfall intensities and low spatial variability. The other two are heavy convective storms with high rainfall intensities, spatial variability and a mixture of both stratiform and convective rainfall.





## 3 Results

### 3.1 Parameterization

In the following, we report on the parameterization of the cascade generator models for the new EVA model and the classical cascade applied to the 100, $1 \times 1$ km$^2$ 5-min radar rainfall snapshots between 2008 and 2018. The procedure used to estimate

the model parameters $a$, $b$ and $c$ for each event is described in Section 2.5. For completeness, two different approaches are considered. In the first, the values of $a$, $b$ and $c$ are estimated using only the coarse-scale data, as one would do in practice. In the second, the values of $a$, $b$ and $c$ are estimated using the high-resolution data at the target scale of $1 \times 1$ km$^2$ (which are unknown in practice). The latter represent the best possible estimates that we can make of the "true" underlying cascade generator parameters and will be used as a reference for assessing the bias in coarser resolution estimates. Table 2 shows the

obtained parameter estimates for the first 4 events in the database for four different spatial resolutions of $1 \times 1$ km$^2$, $2 \times 2$ km$^2$, $4 \times 4$ km$^2$ and $8 \times 8$ km$^2$. It shows large differences between estimated model parameters as a function of the methods and spatial resolutions. Figure 4 gives a more general overview of the problem, showing the $a$, $b$ and $c$ parameters for all 100 fields in the database for an input resolution of $8 \times 8$ km$^2$. The large scatter and low coefficients of determination suggest that it in general, is not possible to reliably infer the true cascade generator directly from coarse scale data (both for the EVA and classical

methods). Specifically, we can see that the $a$ parameter tends to be overestimated while the $c$ parameter is underestimated. For $b$, no systematic bias could be observed. However, the low coefficients of determination of 0.36 and 0.43 suggest that coarse-scale estimates are affected by considerable sampling uncertainty. The fact that $c$ is often zero when estimated from coarse-scale data is a statistical artifact caused by the lack of spatial resolution. It wrongly suggests that the size of a grid cell has no statistically significant effect on the variance of the generator, which is obviously not true as estimates of $c$ obtained using

the high-resolution data are never zero. This is caused by the limited range of variation for $A$ in the coarse-scale data which makes it impossible to correctly estimate the variance of the generator when $A \rightarrow 0$. By contrast, the behavior of the generator when $R \rightarrow 0$ (i.e., the $b$ parameter) is much easier to guess, as both low and large rainfall intensities remain possible even at coarser spatial scales. Comparing the root mean square errors for the EVA and classical cascade models in Figure 4, we see that parameters estimated via the EVA framework tend to be slightly more robust to changes in the input resolution. Nevertheless,

both methods suffer from estimation biases and neither of them is capable of perfectly recovering the true generator model from coarse-scale data, even for relatively modest downscaling ratios (i.e., 64 in this case). Sampling effects obviously play an important role in this, but also the fact that rainfall fields are not perfectly scale invariant. Therefore, the splitting and scaling information retrieved from the coarse scale fields may not reflect what happens at smaller scales or specific areas in the field, especially if the rainfall is highly heterogeneous and intermittent. The conclusion is that in applications involving

downscaling ratios larger than approximately 64, it is generally not possible to retrieve reliable cascade generator parameters directly from coarse-scale data. However, good results might still be possible with the help of climatological generator models or alternatively, by combining multiple successive time steps together to increase sample size and obtain less noisy sample estimates of $\sigma(A, R)$.





Another important observation that can be made concerns the variance of the generator for the EVA and classical models. Figure 5 shows the standard deviation $\sigma(A,R)$ of the empirical breakdown coefficients for all 100 radar snapshots as a function of area $A$ and rainfall intensity $R$. The left column shows the results for the EVA cascade while the classical model is depicted on the right. We see that empirical cascade weights in the EVA model tend to have lower variance compared with the classical

framework (0.409 compared to 0.535), especially for larger values of $A$. This is the consequence of the way grid cells are split in the EVA approach, through integration of the rainfall amount rather than splitting grid cells in two equal parts. Figure 6 illustrates this point by showing the empirical breakdown coefficients $W_1$ and $W_2$ for a $16\times16$ km$^2$ sub-domain belonging to event 1. Since in this case most of the rainfall is concentrated in the left part of the domain, splitting grid cells vertically results in a very uneven redistribution of rainfall rates. In the classical cascade, the left part receives 96.3% of the total rainfall volume

while the right part only receives 3.7% ($W_1 = 0.963$ and $W_2 = 0.037$). The EVA model also produces an uneven split, with half of the total rainfall amount being assigned to an area 82.2% the size of the parent grid cell to the right of the domain while the other half is assigned to the remaining 17.8% ($W_1 = 0.822$ and $W_2 = 0.178$). Overall, however, the EVA split is more balanced. The same conclusion applies to horizontal splits, with the EVA method producing slightly more balanced weights (55.5%-44.5%) than the classical framework (59.4%-40.6%). Of course, in reality, many more grid cells must be taken into

account when calculating the variance of the generator around 0.5. But the key point here is to understand that the generator of the EVA cascade tends to have lower overall variance, making it easier to estimate from a limited number of sample splits. Also, the adaptive sampling strategy in the EVA model reduces sensitivity to the input resolution, resulting in a slightly better power-law fit in Equation (8) (i.e., R$^2$ of 0.66 for EVA versus 0.61 for the classical method). Nevertheless, improvements are not systematic and differences between the two methods can be rather subtle. For very homogeneous rainfall fields for example,

both approaches will essentially be identical, and the classical way of splitting might even be better. But for strongly variable and intermittent fields, the EVA model is likely to provide a significant practical advantage over the classical approach (see next Section).

## 3.2   Visual assessment of downscaled fields

Figure 7 shows some examples of downscaled rainfall fields obtained using the EVA and classical cascade models for the 4 first

events in the database. In all 4 cases, the downscaling ratio was 64. In other words, the original radar rainfall snapshots were first aggregated (i.e., block-averaged) to $8 \times 8$ km$^2$ before being downscaled back to there original resolution of $1 \times 1$ km$^2$. The cascade generator models needed to run the downscaling schemes were estimated directly from the coarse-scale $8 \times 8$ km$^2$ resolution data, as one would do in practice.

Comparing the outputs of the EVA and the classical cascade, one can see that the EVA cascade tends to produce smoother

fields with lower overall variance and peak intensities. Visually, the fields appear to be in better agreement with the original radar snapshots, both in terms of distribution and spatial structure (see Section 3.3 for more quantitative comparisons). Visually speaking, one of the biggest disadvantage of the EVA cascade appears to be the fact that the resulting fields look slightly blocky, with some of the initial coarse-scale grid cells still visible. The blockiness can be attributed to biased parameter estimates $a$, $b$ and $c$ caused by the limited range of spatial scales available for studying the splitting behavior of grid cells. In particular,





the previous section has shown that the $c$ parameter which controls the splitting of grid cells with respect to area tends to be underestimated when derived from coarse scale data, causing the cascade to converge too quickly. The classical model does not appear to produce these blocky patterns. On the contrary, downscaled fields appear to be too variable compared with the observations. Again, the discrepancies can be attributed to biased cascade generator parameters. But in this case,

the main problem appears to be the strongly overestimated $a$ parameter which controls the overall variability of the splits and compensates for the underestimated $c$ parameter. As shown by these 4 examples, none of the downscaled methods appears to be able to perfectly reproduce the small-scale properties of the underlying rainfall field. However, the fact that one method tends to underestimate the total variability while the other tends to overestimate it is interesting. It highlights the complementary nature of the two approaches and perhaps, could be exploited during further post-processing steps and/or quality control steps.

Before moving on to more quantitative assessments, there is another important point that needs to be made here concerning the individual performances of the two random cascade models. The problem with Figure 7 is that it only shows the performance of the two cascade models for the sub-optimal cascade generators estimated from coarse-scale data. While this might be representative of the actual performance in real-life conditions, it is not really a fair comparison of the two methods. Indeed, a large part of the differences between EVA and the classical cascade in Figure 7 can be attributed to the biased cascade

generator parameters, and not the model itself. Therefore, to compare the two methods on a truly fair basis, one also needs to say something about the performance under optimal conditions (i.e., unbiased parameter estimates). To do this, additional experiments were performed in which the same 4 rainfall fields were downscaled with the help of the best possible generator model derived from the $1 \times 1$ km$^2$ data (see Figure 8). Comparing Figure 7 to Figure 8, we can see a big improvement in the performance of the classical cascade model which becomes comparable (at least visually) to the EVA cascade. The conclusion

is that both models are capable, in theory, of producing similarly good results. However, since the true cascade generator model is likely to be unknown in practice, the more robust EVA cascade is expected to stay much closer to the optimal performance, providing a clear advantage in situations where the generator must be estimated from the coarse-scale data.

## 3.3  Quantitative assessment of downscaled fields

We continue the assessment of the downscaled fields by studying the probability distribution functions of rainfall rates gener-
ated by the random cascades. Figure 9 shows the quantiles of observed and downscaled rainfall rates for the 4 first events and a downscaling ratio of 64 ($8 \times 8$ km$^2$ to $1 \times 1$ km$^2$, 100 random realizations for each event). Each cascade model is represented by two boxplots: the first shows the quantiles of rainfall rate obtained when the generator is derived from the coarse-scale data while the second shows the results for when the generator is derived using the $1 \times 1$ km$^2$ data. The second generator is unknown in practice but provides further insight into the sensitivity of the performance to parameterization issues. It also gives a good
idea of the best possible achievable performance for each model. Finally, we also provide the quantiles obtained when applying bilinear interpolation which is well-known for producing fields that are too smooth compared with the observations, strongly underestimating higher quantiles and small-scale extremes.

We start the analysis by looking at the rainfall rates generated by the classical random cascade model. Overall, the distributions appear to be in relatively good agreement with the observations. However, some important discrepancies remain,



especially for the very high quantiles. Performance appears to be very sensitive to parameterization issues, varying a lot depending on the generator and from one event to another. Homogeneous, low intensity events such as event 2 are reproduced rather well. But in events 1 and 4, extremes are clearly overestimated. In fact, in the majority of the 100 considered events, the classical cascade overestimates rainfall extremes when the coarse-scale generator is used. However, there are also a few inter-

esting exceptions to this rule. For example, in event 3, the classical cascade underestimates the 99.9% quantile compared with the observations. The problem with event 3 is that the rainfall field is highly heterogeneous, consisting of multiple convective and stratiform areas of different sizes, shapes and orientations. Therefore, big local differences in scaling behavior exit within the field, making it hard to derive a meaningful cascade generator model that applies to the entire domain. This is highlighted by the fact that the coarse-scale generator actually produces better result than the fine-scale generator, which is highly unusual

and points to serious problems during parameter estimation.

We now turn our attention to the EVA model. Objectively, there appears to be no substantial improvement in terms of the model's ability to reproduce higher rainfall rates. One advantage is that the outcomes of the EVA cascade are more predictable and consistent compared to the classical approach. However, the produced distributions are clearly too narrow compared with the observations and the model underestimates higher rainfall quantiles and small-scale extremes. Still, the underestimation is

much less severe than for bilinear interpolation. The systematic underestimation of higher rain rates is a problem but can be explained by the fact that the variance of empirical EVA cascade weights for small values of $A$ tends to be slightly underestimated due to the additional interpolation step (see Section 2.5 for more details). Figure 10 provides more insight into this by showing the empirical semivariance values of rainfall intensities for distances of 1 km up to 8 km (i.e., the sub-grid variability generated during the downscaling). It confirms that the EVA cascade produces fields that are slightly too smooth while the clas-

sical cascade tends to overestimate small-scale variability. Moreover, the ensemble spread for the EVA model is slightly lower than for the classical cascade. This means that for a fixed cascade generator model, the EVA model produces rainfall fields with almost identical distributions and spatial structures. Individual realizations may still look different on a pixel-by-pixel basis, but their average statistical properties (e.g., histograms and variograms) will be almost identical. This can be an advantage but also means that in order to produce truly representative ensembles that capture a large enough range of possible scenarios, it is

better to run the EVA cascade several times with slightly perturbed model parameters $a$, $b$ and $c$ rather than generating a large number of fields with the same generator.

Figure 11 gives a broader overview of the performance over the 100 selected events for a downscaling ratio of 64 and coarse-scale sample generator. It confirms what we have seen before, namely that the classical cascade model tends to overestimate high rainfall rates while the EVA model tends to underestimate them. Nevertheless, the higher coefficient of determination $R^2$

between observations and downscaled rainfall rates and the better agreement in terms of reproduced semivariance values show that the new EVA cascade model tends to outperform the classical approach, both in terms of the reproduced spatial correlation structure but also in terms of its ability to reproduce consistent small-scale extremes. In both cases, systematic biases remain which were attributed to difficulties in getting reliable generator estimates from coarse-scale data. Also, performance clearly decreases with intermittency (see Figure 11c-d), which can be explained by the fact that fewer samples are available for

estimating the generator but also by the fact that highly intermittent fields tend to be more heterogeneous, with larger local





deviations from scale-invariance than their homogeneous counterparts. Because it is more robust to sampling uncertainty, the EVA model tends to produce more reliable results in those difficult cases. However, improvements are not systematic and many issues remain. In particular, more development is needed to overcome the drop in performance at intermittency levels above 60% and as well as to mitigate the underestimation of small-scale rainfall extremes which is a fundamental requirement in

downscaling for hydrological applications (Molnar and Burlando, 2005).

Finally, we turn out attention to the performance of the cascade models as a function of the downscaling ratio. Figure 12 shows the 10%, 25%, 50%, 75% and 90% quantiles of the coefficient of determination $R^2$ between observed and downscaled rainfall rates for three different downscaling factors (i.e., 4, 16 and 64). Figures 12a-b show the performance for the coarse-scale sample generator while Figures 12c-d show the best possible performance for the generator derived from $1\times1$ km$^2$

data (unknown in practice). The values corresponding to Figures 12a-b are given in Table 3. We can see that in practical applications where the generator must be estimated from the coarse-scale data, the EVA model outperforms the classical cascade across all three downscaling ratios. As expected, differences between the two methods increase as we move towards larger ratios. However, the EVA model tends to remain much closer to the best theoretical achievable performance than the classical cascade. Again, the small differences between Figures 12c-d confirm that in theory, both cascade models are capable

of achieving a similarly good performance provided that the optimum generator model can be guessed from the data. Even so, the EVA model still appears to have a slight edge over the classical approach, with median $R^2$ values of 0.94, 0.83 and 0.54 against 0.93, 0.81 and 0.52 for the classical method which makes sense given that even the "best" generator model at $1\times1$ km$^2$ was inferred from a limited number of samples and might therefore still be slightly biased. Unfortunately, the relatively small domain size of $128\times128$ km$^2$ meant that no reliable estimates of the generator could be obtained for an input resolution of

$16\times16$ km$^2$ or higher. However, this is an issue related to the choice of the domain size in this study rather than a theoretical limit on the maximum downscaling ratio. Additional experiments on larger domains (not shown here) suggest that decent results can still be obtained for downscaling ratios up to about 256, making the technique applicable to satellite data or global numerical weather models with grid sizes up to ten kilometers. However, accuracy drops rapidly and large uncertainties are to be expected for such large downscaling ratios.

**4 Discussion**

While this research mainly focused on the description of the EVA cascade model, the underlying generator and its application to a few selected case studies, there are numerous complementary research lines that can be pursued. One of them revolves around possible ways to overcome biases in cascade generator parameters and correct for systematic errors as a function of the intermittency and downscaling ratio. Diagnostic tools for detecting potentially problematic cases based on plausible ranges

for each parameter need to be developed. Alternatively, one could apply both an EVA and a classical cascade and compare the obtained results. If they are wildly inconsistent, the EVA model is likely to be closer to the truth. Another possibility would be to design flexible climatological generator values that can be adjusted depending on rainfall type and large-scale properties (e.g., intensity, intermittency, range), an approach that may be more flexible while limiting sampling issues. Preliminary work





performed within this study (not shown) suggests that this may be promising as cascade parameters often tend to be correlated with each other or to large-scale rainfall properties (Guntner et al., 2001; McIntyre et al., 2016).

The second point that is worth discussing concerns the complementary nature of the EVA framework compared with the classical representation in terms of intensity over fixed grid cell sizes. The main advantage of the EVA framework lies in its

adaptive sampling strategy. By flipping the problem around and focusing on the areas for fixed amounts of water rather than the opposite, additional insight into spatial variability of rainfall within grid cells can be gained. Most importantly, occurrence and intensity are not viewed separately anymore but combined together into a single continuous process. All quantities are strictly positive which reduces model complexity, improves the scaling and lowers sampling uncertainty. If rainfall fields were perfectly homogeneous and the sensors used to measure them had unlimited precision, the two representations would be

equivalent. However, since rainfall fields can be highly variable in space and time, and measurements are affected by sampling uncertainties, one of the two representations is likely to be more appropriate or useful in practice. A better understanding of these cases and how to choose the best framework depending on sampling resolution, intermittency and measurement accuracy is key to improving our understanding of the space-time variability of rainfall and its representation in models.

The third issue that needs to be mentioned relates to the assumption that the cascade generator model is location invariant

(i.e., that the same splitting rules apply to all pixels, independently of their location). This is not necessarily valid in highly heterogeneous fields, as highlighted by the poor performance and inconsistent behavior of the cascade models during event 3. The key point here is that there might be specific areas within a domain where rainfall scales radically differently. For example, as we enter individual rainfall cells, spatial variability might change very drastically, with rainfall rates exhibiting abnormally high or almost no variation at all with spatial resolution. At the same time, other elements belonging to larger-scale structures

might evolve together in a more coherent and predictable way than expected based on their size and intensity. One possible solution to overcome this problem would be to define multiple local generators instead of a single universal one. But this is a very challenging problem and more research into algorithms capable of detecting strong local deviations in scaling properties is needed to help pinpoint problematic regions and come up with a better approach. Also, the use of multiple generators would require additional model parameters, which may not necessarily be desirable and should only be considered when absolutely

necessary (e.g., for example to account for strong orographic effects).

The fourth point of discussion concerns possible extensions of the EVA model. Similarly to classical multiplicative random cascades, the EVA cascade can be applied to downscale time series, spatial and space-time data. For time series, the equivalent formalism is give by the notion of "equal-volume times", also known as inter-amount times (Schleiss and Smith, 2016; Schleiss, 2017). Future work will be directed at combining inter-amount times and EVAs to design more general downscaling schemes

for space-time data. Another interesting and possible extension concerns the possibility to include spatial anisotropy into the downscaling process. One way to do this is by using two different generator models, one for vertical and another for horizontal splits. For example, an additional model parameter characterizing the ratio between the standard deviation of the generator for H and V split could be introduced. More generally, one could also define a full set of different model parameters $(a_H, b_H, c_H)$ and $(a_V, b_V, c_V)$ for each type of split. Grid cells could also be rotated and realigned along the principal direction of variability,

allowing for splits along other spatial directions than H and V. This could help in case of highly elongated rainfall cells.





One last point that is worth mentioning concerns the computational complexity of the EVA model. One crucial difference between the EVA and the classical cascade is that the classical stops as soon as the target resolution has been reached. The EVA cascade on the other hand tends to run over more levels, producing many grid cells that are smaller than the target resolution. The total number of cascade levels and grid cells depends on 1) the initial rainfall volumes contained in the coarse scale

grid cells and 2) the bucket capacity prescribed by the user. This means that for large rainfall fields (e.g., several hundreds of km) with high rainfall intensities, the number of generated grid cells can be in the order of several millions. As a result, both runtime and memory usage will be larger than for a classical cascade. However, there are various ways to limit the computational burden. The easiest is to stop splitting grid cells once they are about 3-4 times smaller than the target resolution regardless of how much water they contain. Similarly, grid cells that are entirely contained within a target resolution pixel do

not need to be split up further (regardless of their size and amount), as these additional splits would not be visible after the re-sampling anyway. Similarly, there is no need to split up grid cells once they have converged to a fixed rainfall intensity, i.e., when $\sigma(A, R) \approx 0$, as this would only result in a higher number of sub-grid cells with identical intensities and would not add any new information. The obvious downside to these numerical tricks is a loss in flexibility, as users need to decide on a fixed target resolution before running the cascade.

## 5   Conclusions

A new multiplicative random cascade for downscaling intermittent rainfall fields based on the concept of equal-volume areas (EVA) has been proposed. Downscaling experiments on 100 high-resolution radar rainfall snapshots in the Netherlands have shown that on average, the EVA cascade outperforms its competitors, both in terms of the reproduced rainfall distributions and spatial structures. Improvements are mainly attributed to the adaptive sampling strategy in the EVA formalism which

avoids zero rainfall values and leads to more accurate and robust model estimates in the presence of intermittency. The new proposed logit-normal cascade generator model with scale and intensity-dependent variance ensures that every grid cell in the EVA cascade eventually converges to a fixed intensity or a fixed area, putting the new model in the category of bounded microcanonical cascades. Despite the encouraging results, improvements are not systematic and many challenges remain. The most important is that the EVA cascade tends to underestimate small-scale extremes, producing fields that are slightly too

smooth and blocky compared with the observations. This is attributed to biased model parameters and more generally, to the difficulty of retrieving the true cascade generator from coarse-scale data. The fact that cascade weights in the EVA framework must be estimated using linear interpolation is also a clear weakness, causing $\sigma(A, R)$ to be underestimated when $A \to 0$ (i.e., small grid cells tend to split too evenly). On the other hand, one also needs to be aware of the fact that the classical cascade model based on fixed grid cells suffers from the opposite problem, strongly overestimating the small-scale variability

and magnitude of rainfall extremes. The complementary nature of the two approaches and the fact that they tend to produce opposite errors opens new possibilities for quality control and bias corrections of downscaled fields.

Apart from introducing a new model, the present study also clearly highlighted the outstanding challenges associated with downscaling intermittent rainfall fields. The most important issue concerns the estimation of cascade generator models from





coarse-scale data. Sensitivity analyses performed within the framework of this study clearly showed that two of the cascade model parameters (i.e., $a$ and $c$) tend to be biased when estimated from coarse-scale data. However, the EVA model seems to be more robust to these sampling issues. This is the single most important advantage of the EVA model compared with the classical approach and also the main factor responsible for the higher performance. However, it is also worth mentioning

that in principle, good performance remains possible for both cascades models for downscaling ratios up to 128-256, provided that the optimal cascade generator can be guessed from the data. While interesting from a theoretical point of view, this last result may be of limited usefulness in practice as the optimal generator is likely to be unknown. Also, for large domains of several hundreds of kilometers and highly heterogeneous fields, it might not always be possible to adequately describe the complex redistribution of water across scales using a single location-invariant cascade generator. Event 3 is a good example

of such a case, with both cascade models struggling to reproduce realistic small-scale patterns. However, one can always tune performance to specific events and gradually improve the agreement by introducing more model parameters if necessary. But performance is not everything and the EVA model offers many other advantages that are worth investigating in more detail. Therefore, it should not be seen as a new model designed to replace traditional cascades based on intensity but rather as a new complementary tool for modelers to get new insight into the complex spatio-temporal organization of rainfall across scales and

to design efficient, easy to parameterize and robust solutions for downscaling intermittent fields.

*Competing interests.*   The author has no competing interests.

*Acknowledgements.*   Funding for this work was provided through 'Water JPI Europe', ERA-NET Co-fund WaterWorks2014 project MUFFIN (Multiscale Urban Flood Forecasting: From Local Tailored Systems to a Pan-European Service) and through the Netherlands Organisation for Scientific Research NWO (project code ALWWW.2014.3). Special thanks to the Royal Netherlands Meteorological Institute (KNMI) for

collecting and providing the radar data used in this study.



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





**Figure 1.** Schematic of the branching rules for the classical and equal-volume area (EVA) random cascades. The area is denoted by $A$ and the rainfall volume by $V$. The random weights are $W_1, W_2, \ldots, W_6$.





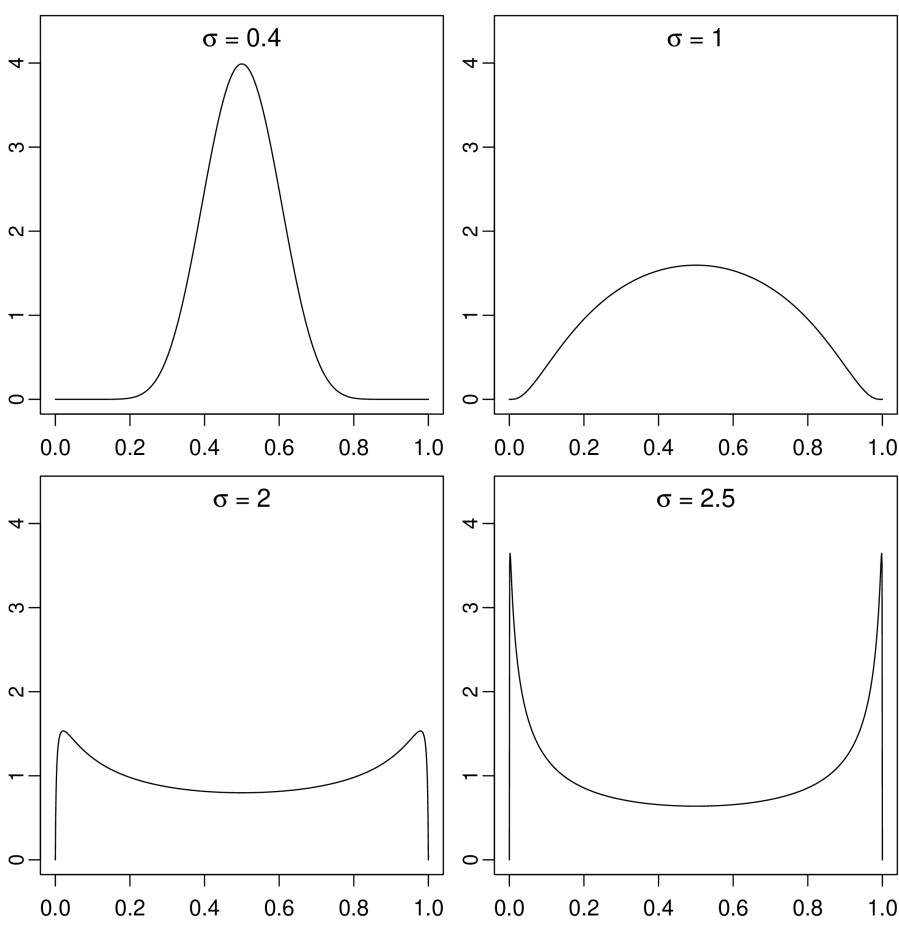

**Figure 2.** Theoretical distribution of the logit-normal cascade weights $W$ in Equation (7) for $\mu = 0$ and different values of standard deviation $\sigma$.

**Figure 3.** Original 1×1 km$^2$ and upscaled (4×4 km$^2$ and 8×8 km$^2$) 5-min radar rainfall snapshots for events 1 to 4.



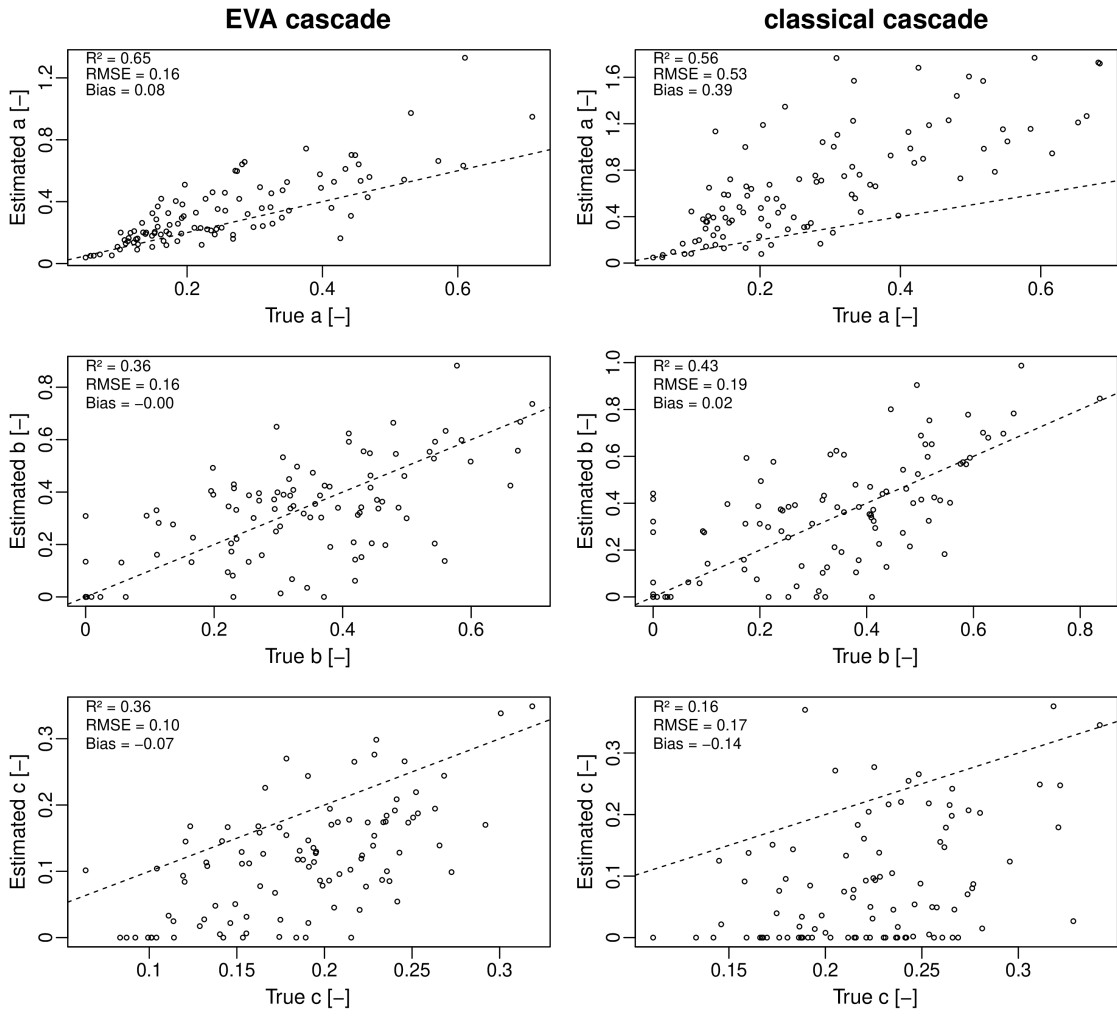

**Figure 4.** Estimated coarse-scale generator parameters $a$, $b$ and $c$ for an input resolution of $8 \times 8$ km$^2$ versus the fine-scale parameter values derived using the $1 \times 1$ km$^2$ data for the 100 selected events.

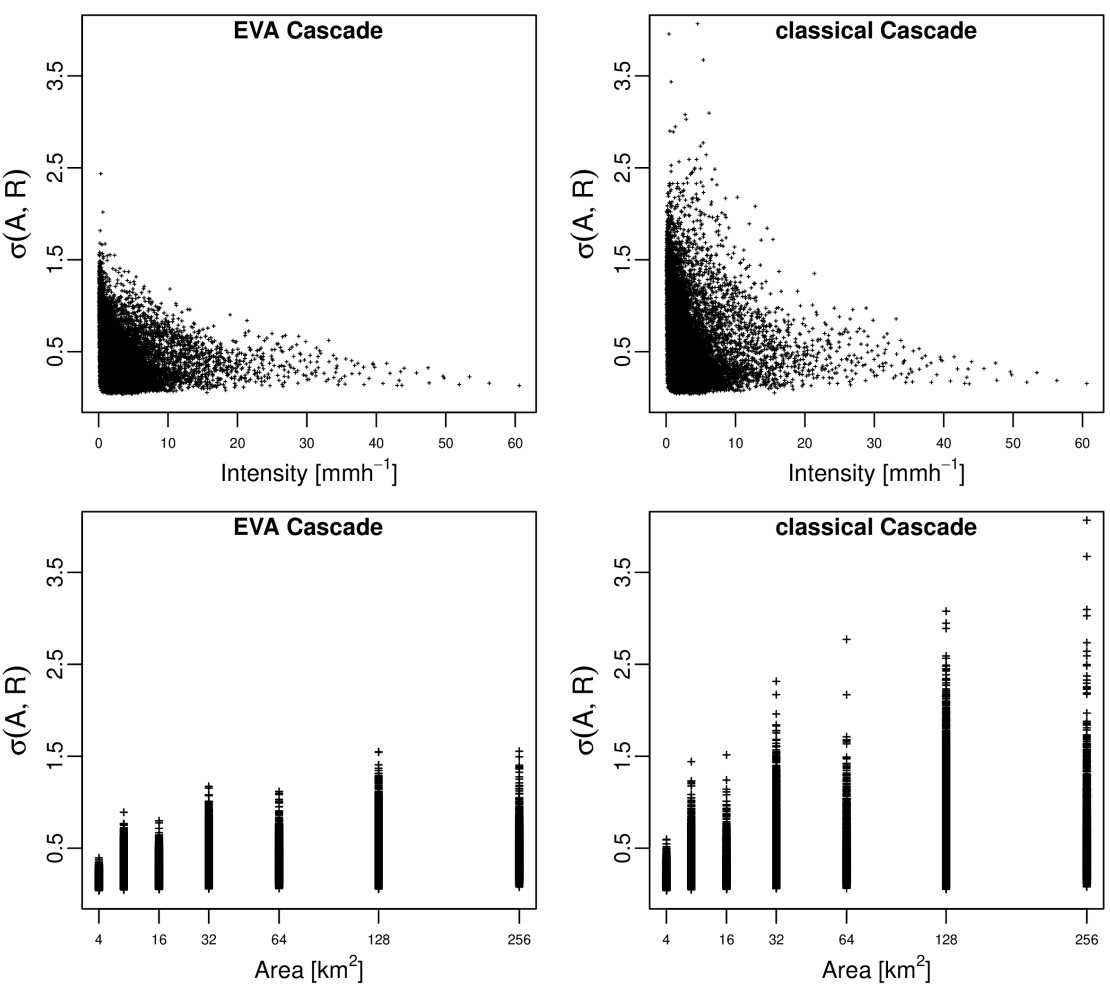

**Figure 5.** Standard deviation of empirical breakdown coefficients for the 100 radar snapshots in the database as a function of the rainfall intensity $R$ and area $A$ of grid cells.



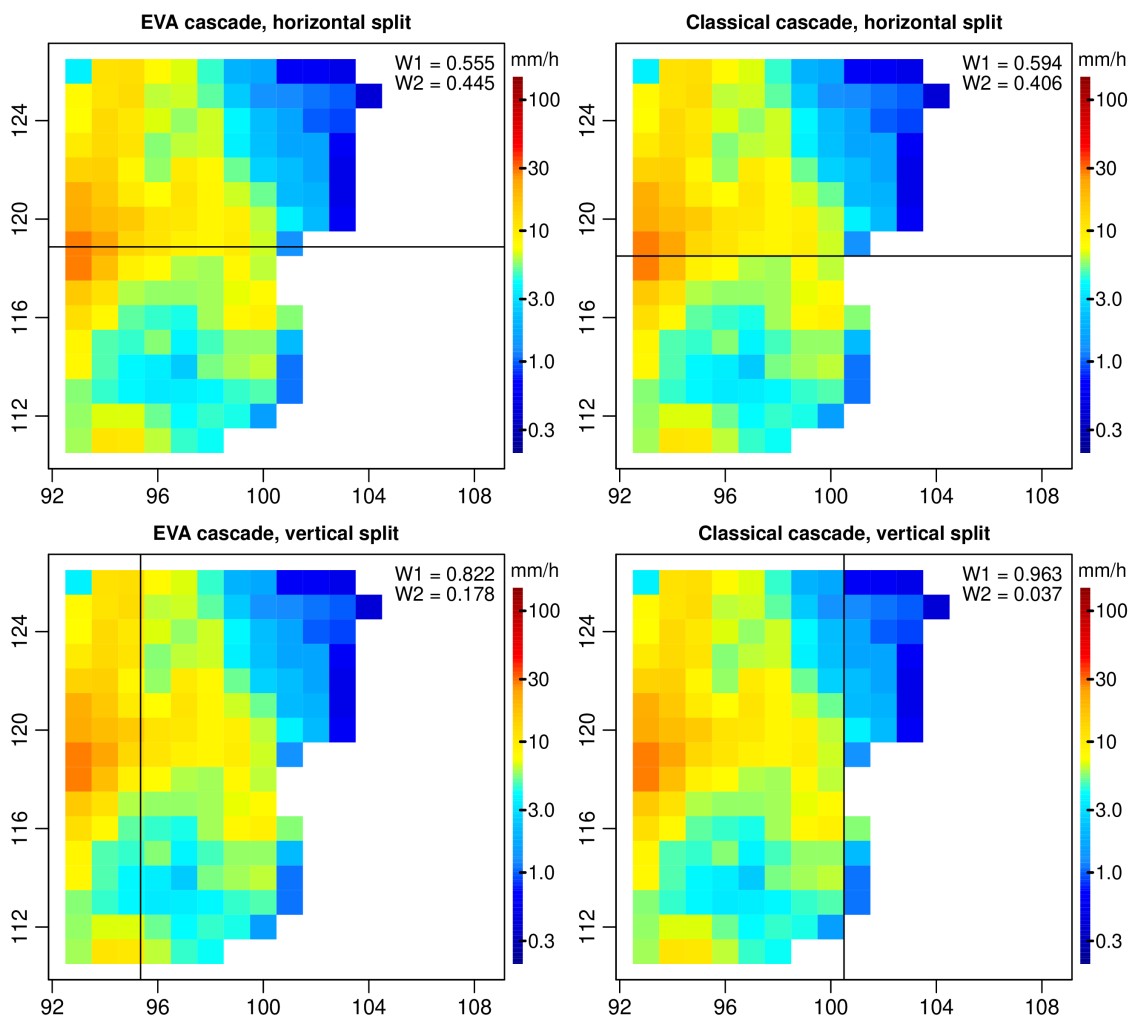

**Figure 6.** Example of empirical breakdown coefficients $W_1$ and $W_2$ for a 16×16 km² grid cell in event 1 (convective). The splits corresponding to the EVA model are shown on the left. The ones for the classical model are shown on the right.

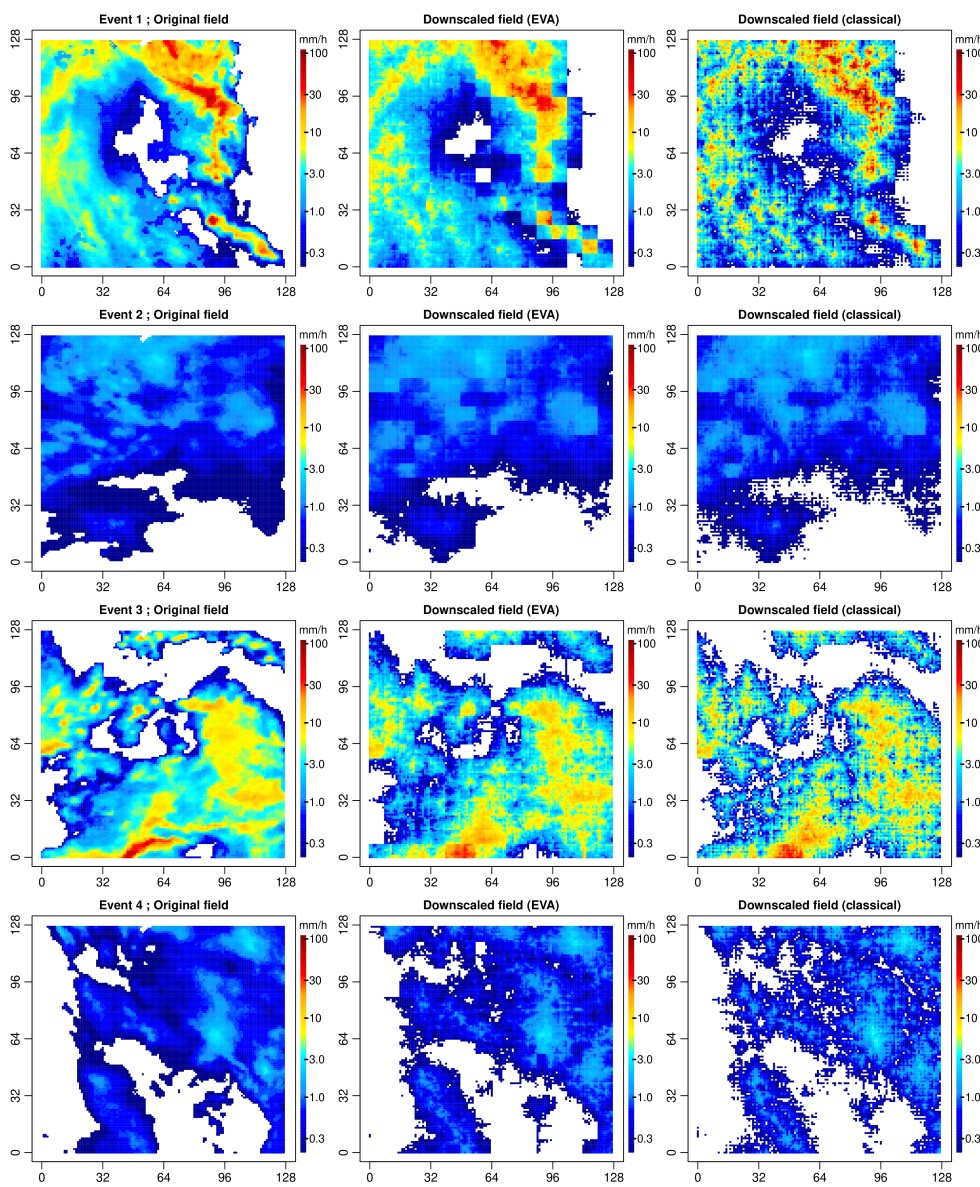

**Figure 7.** Downscaled rainfall fields for events 1-4 and a downscaling factor of 64 (i.e., input resolution of $8 \times 8$ km$^2$ and target resolution of $1 \times 1$ km$^2$). The left column shows the original radar rainfall snapshots at $1 \times 1$ km$^2$. The middle and right columns show the outputs of the EVA and classical cascade models for the (biased) coarse-scale sample generator. Only the first of 100 different random realizations for each field and cascade model is shown.



**Figure 8.** Downscaled rainfall fields for events 1-4 and a downscaling ratio of 64 (i.e., input resolution of $8\times8$ km$^2$ and target resolution of $1\times1$ km$^2$). Similar format to Figure 7 except that the generator model was derived from the $1\times1$ km$^2$ data.



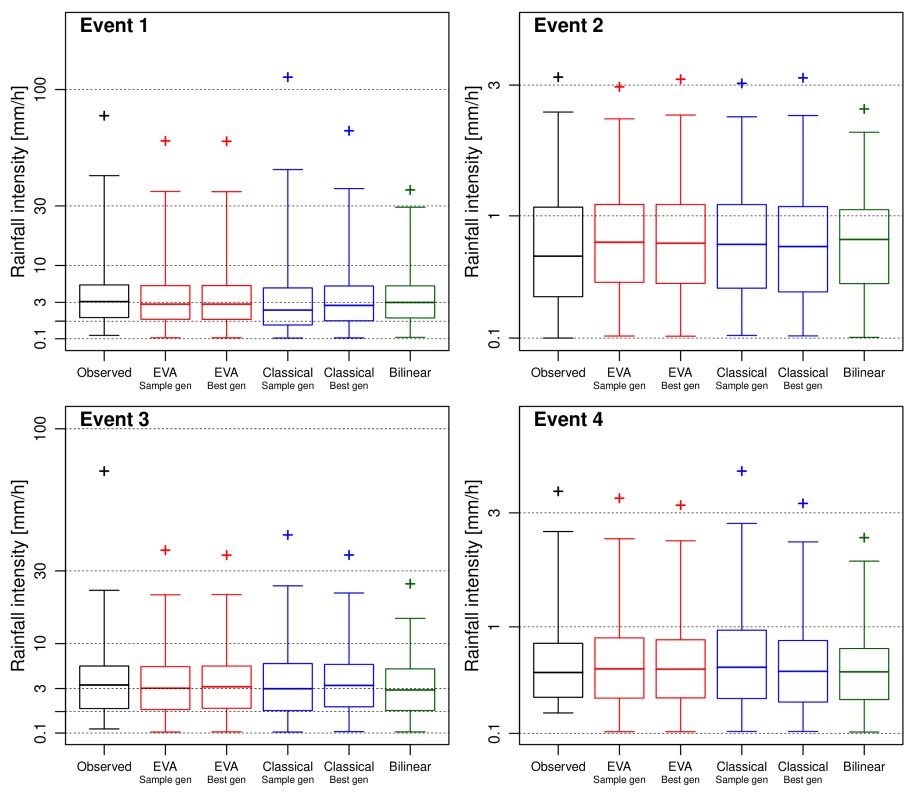

**Figure 9.** Observed versus downscaled rainfall rates for the first 4 events in the database and a downscaling ratio of 64 (i.e., input resolution of 8×8 km$^2$ and target resolution of 1×1 km$^2$). The boxplots denote the 1%, 25%, 50%, 75% and 99% quantiles of rainfall rates (given occurrence). The crosses represent the 99.9% quantiles among 100 different random realizations.

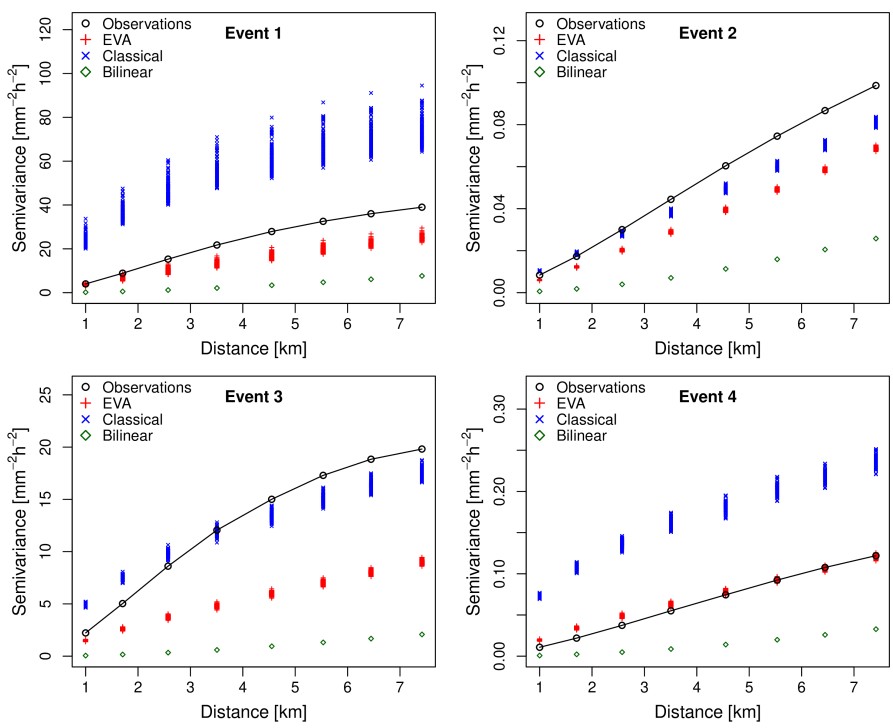

**Figure 10.** Sample variograms of rainfall intensity (given occurrence) for events 1-4 and spatial displacements up to 8 km. The downscaling factor is 64 (i.e., input resolution of 8×8 km$^2$ and target resolution of 1×1 km$^2$). For each cascade model, 100 different realizations were generated. The generator model was estimated from the coarse-scale data at 8×8 km$^2$ resolution.

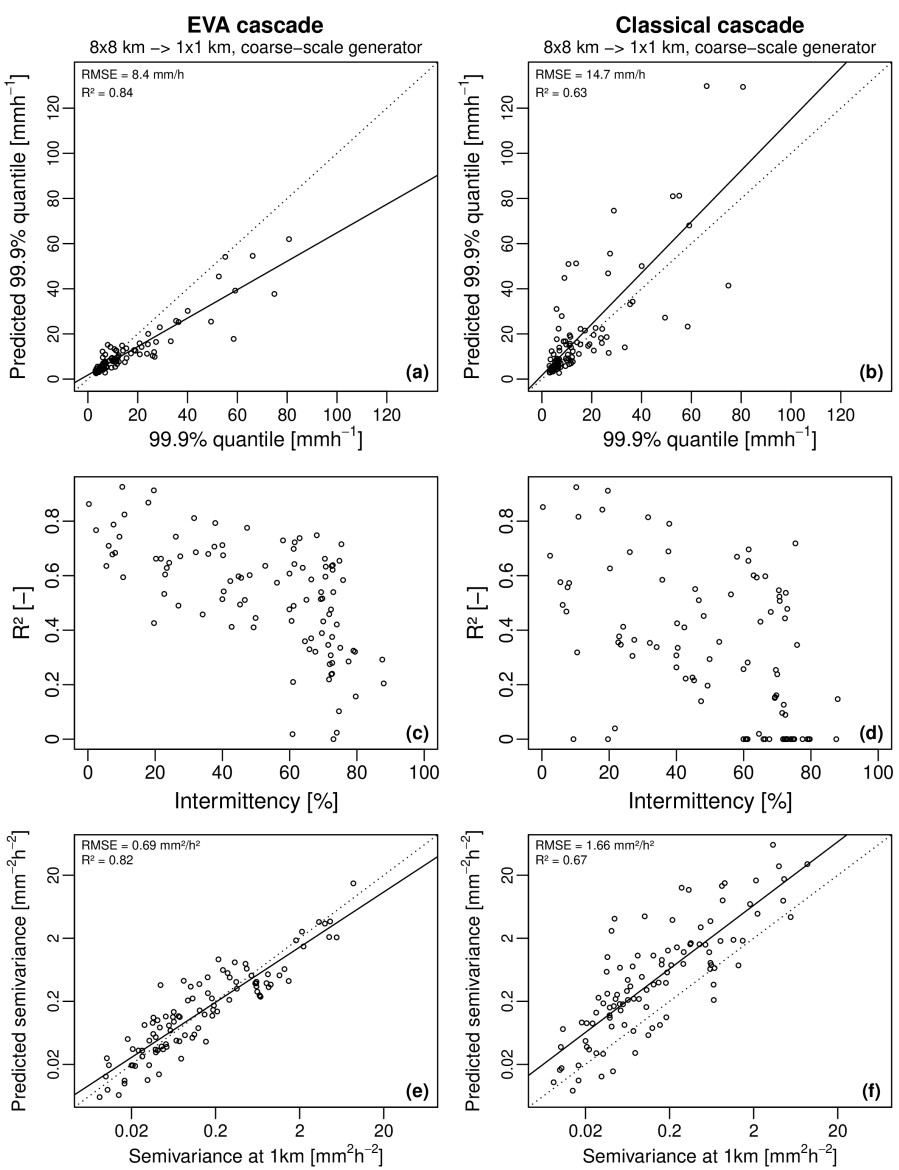

**Figure 11.** Overall performance of the random cascade models for 100 high-resolution radar rainfall fields, coarse-scale sample generator estimate and downscaling factor of 64 (i.e., input resolution of $8 \times 8$ km$^2$ and target resolution of $1 \times 1$ km$^2$). (a)-(b) show the predicted versus observed 99.9% quantile of rainfall intensity, (c)-(d) the coefficient of determination $R^2$ between downscaled and observed rainfall rates as a function of intermittency (i.e., the fraction of zero rainfall values in the domain) and (e)-(f) show the predicted versus observed semivariance values for a 1 km spatial displacement. The EVA cascade is shown on the left and the classical cascade on the right.

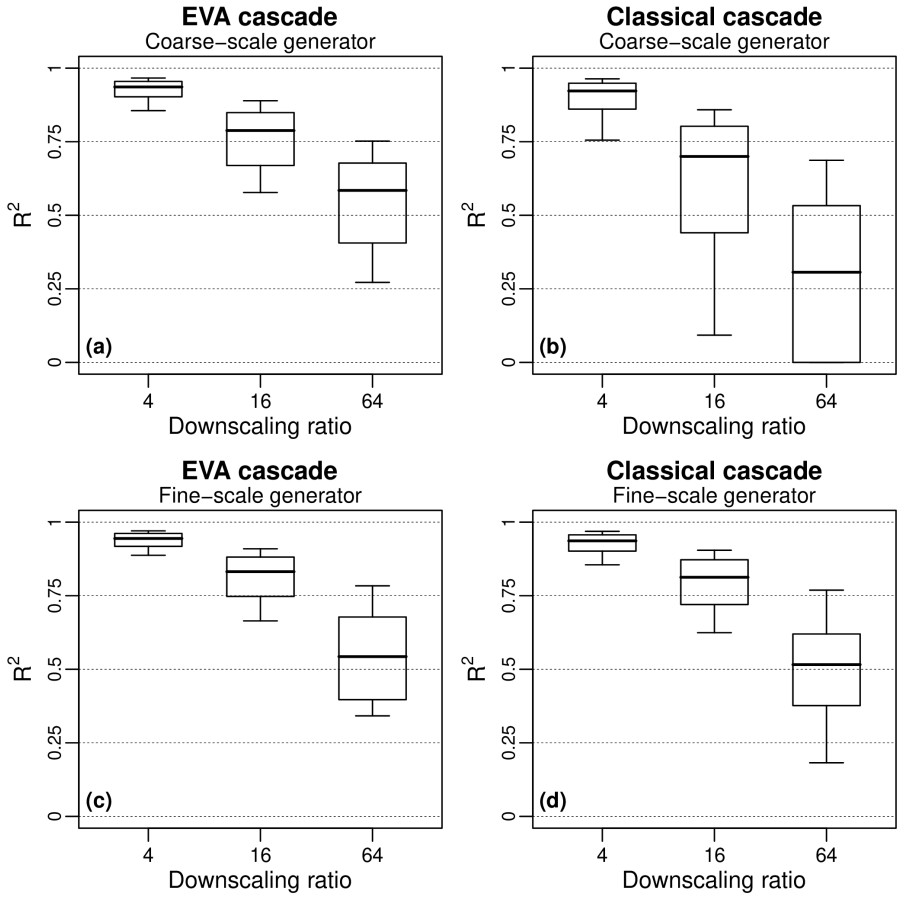

**Figure 12.** 10%, 25%, 50%, 75% and 90% quantiles of the coefficient of determination $R^2$ between observed and downscaled rainfall fields for the 100 selected rain events. The values corresponding to the coarse-scale generator are given in Table 3.





**Table 1.** Summary statistics for the 4 example events: time, proportion of zero rainfall values $p_0$, average rainfall intensity $\bar{R}^+$ (given occurrence), maximum rainfall intensity $R_{\max}$, variance $\sigma^2_{R+}$ of rainfall intensity (given occurrence) and spatial decorrelation range of the rainfall intensity field (given occurrence).

| Event | Time | $p_0$ [%] | $\bar{R}^+$ [mmh$^{-1}$] | $R_{\max}$ [mmh$^{-1}$] | $\sigma^2$ [mm$^2$h$^{-2}$] | Range [km] |
|---|---|---|---|---|---|---|
| 1 | 2009-05-26 02:50 | 21.7 | 5.37 | 97.2 | 65.2 | 23.4 |
| 2 | 2015-11-09 16:55 | 18.0 | 0.77 | 3.7 | 0.35 | 73.3 |
| 3 | 2009-11-23 17:05 | 22.7 | 4.51 | 89.2 | 30.3 | 26.3 |
| 4 | 2009-12-08 02:40 | 32.1 | 0.65 | 4.1 | 0.24 | 33.8 |





**Table 2.** Model parameter estimates $a$, $b$ and $c$ for the first 4 events for input resolutions of $1\times1$ km$^2$, $2\times2$ km$^2$, $4\times4$ km$^2$ and $8\times8$ km$^2$

| | $1\times1$ km$^2$ | | | $2\times2$ km$^2$ | | | $4\times4$ km$^2$ | | | $8\times8$ km$^2$ | | |
|---|---|---|---|---|---|---|---|---|---|---|---|---|
| | a | b | c | a | b | c | a | b | c | a | b | c |
| event 1, EVA | 0.17 | 0.02 | 0.23 | 0.22 | 0.09 | 0.21 | 0.37 | 0.22 | 0.12 | 0.19 | 0.00 | 0.17 |
| event 1, classical | 0.19 | 0.01 | 0.26 | 0.21 | 0.05 | 0.26 | 0.59 | 0.11 | 0.08 | 0.64 | 0.00 | 0.05 |
| event 2, EVA | 0.07 | 0.31 | 0.27 | 0.08 | 0.39 | 0.22 | 0.07 | 0.44 | 0.24 | 0.06 | 0.53 | 0.24 |
| event 2, classical | 0.08 | 0.36 | 0.27 | 0.09 | 0.44 | 0.22 | 0.08 | 0.49 | 0.24 | 0.10 | 0.61 | 0.20 |
| event 3, EVA | 0.40 | 0.43 | 0.16 | 0.58 | 0.48 | 0.08 | 0.66 | 0.40 | 0.04 | 0.49 | 0.34 | 0.08 |
| event 3, classical | 0.59 | 0.56 | 0.16 | 0.86 | 0.52 | 0.06 | 1.17 | 0.46 | 0.00 | 1.16 | 0.40 | 0.00 |
| event 4, EVA | 0.10 | 0.44 | 0.24 | 0.15 | 0.42 | 0.16 | 0.21 | 0.44 | 0.08 | 0.20 | 0.46 | 0.09 |
| event 4, classical | 0.10 | 0.50 | 0.26 | 0.16 | 0.46 | 0.17 | 0.31 | 0.49 | 0.05 | 0.44 | 0.52 | 0.00 |





**Table 3.** 10%, 25%, 50%, 75% and 90% quantiles of the coefficient of determination $R^2$ between observed and downscaled rain rates for the EVA and classical method and three different downscaling factors (coarse-scale sample generator only).

|      | $2{\times}2$ km$^2$ |           | $4{\times}4$ km$^2$ |           | $8{\times}8$ km$^2$ |           |
|------|------|-----------|------|-----------|------|-----------|
|      | EVA  | Classical | EVA  | Classical | EVA  | Classical |
| 10%  | 0.86 | 0.76      | 0.58 | 0.09      | 0.27 | 0.00      |
| 25%  | 0.90 | 0.86      | 0.67 | 0.44      | 0.41 | 0.00      |
| 50%  | 0.94 | 0.92      | 0.79 | 0.70      | 0.58 | 0.31      |
| 75%  | 0.95 | 0.95      | 0.85 | 0.80      | 0.68 | 0.53      |
| 90%  | 0.97 | 0.96      | 0.89 | 0.86      | 0.75 | 0.69      |