# Peer review of "A new discrete multiplicative random cascade model for downscaling intermittent rainfall fields"

_Hydrology and Earth System Sciences, 2019_

## Referee Comment (RC1) · Anonymous Referee #1 · 5 Dec 2019

In this paper the new approach to downscale precipitation field, based on equal-volume areas (EVAs) is evaluated.

The topic of the paper is a relevant issue for hydrology, meteorology and water management.

The author compered EVA method and "classical" discrete multi-canonical random cascade and bilinear interpolation. The composition of the paper is valid. The methods are clearly described. The author mentioned both pros and cons of the EVA cascade generator and pointed the directions of further development.

The results of EVA method application are worth to be published. However, some

details should be corrected:

1. Page 6 line 30: "(. . .) from uniform (Olsson, 1998) to log-normal, Beta (. . .)" - Please add reference to log-normal distribution usage and correct spelling to "beta".

2. Page 9 line 6: "While the EDA (. . .)" - Please check spelling.

3. Page 10 line 11: "It shows large differences between estimated model parameters as a function of the methods and spatial resolutions." - No relationship in tab. 2 is observed. Please paraphrase this sentence.

4. Figure 4 - Please use the same bounds in vertical axis for the same parameter and do not use "true" adjective in horizontal axis title. Please name axis explicitly – which parameter value is obtained for coarse-scale generator and which for radar data resolution.

5. The author used plural form "we" multiple times, however, he is the only person sign to this paper.

6. Figure 9 - Please correct the labels: "Sample gen" and "Best gen" – it is not known which is for 8x8 km resolution and which is for 1x1 km resolution.

7. Page 13 line 7: "Therefore, big local differences in scaling behavior exit within the field" - Please check spelling.

8. Page 13 line 33: "Also, performance clearly decreases with intermittency" - Please add if the intermittency was calculated for empirical precipitation fields or for generated ones.

9. Page 14 line 23: "However, accuracy drops rapidly and large uncertainties are to be expected for such large downscaling ratios." - Please check grammar.

10. Page 14 line 23: "(. . .) the EVA model is likely to be closer to the truth." - Please paraphrase this part of the sentence (the model results where compared with weather radar scans) - for instance "(. . .) the EVA model is likely to be closer to the observed

precipitation fields".

---

## Referee Comment (RC2) · Elena Volpi (Referee) · 17 Dec 2019

**1    Summary**

The manuscript proposes a new approach for rainfall disaggregation, based on the framework of Multiplicative Random Cascade (MRC) yet able to overcome the its limitations, namely possible over-parametrization or limited accuracy in presence of intermittency (more specifically, for a large amount of zero values). The Author investigates the effectiveness of a disaggregation scheme based on the notions of scaling and Equal Volume Areas (EVA), with maximum three parameters that can be inferred

directly from the coarse-scale data. Investigation is performed for 2D (spatial) rainfall fields. The proposed approach allows for intermittency simulation without explicitly coding zero values and their distribution and structure; on the other hand, the simulated rainfall field needs to be re-sampled in grid cells from variable to constant size (i.e., re-gridding).

**2  General comment**

The manuscript is well written and organized, and the topic is surely of interest for the hydrologic community. Hence, I suggest to consider the manuscript for publication in HESS. Before this, more details are necessary to allow for reader understanding and repeatability of the work (that is also the application of the framework in different contests). The most important problem is that the theoretical properties of the simulated process, that are the marginal probability distribution function and the joint distribution, are not explicitly presented and discussed; these are fundamental to understand e.g. if the model is based on the same scaling property ruling the theoretical behavior of the traditional MRC. I have some additional concerns that are listed below. I hope they will be helpful for manuscript improvement.

**3  Specific and technical comments**

- I'm curious about the reasons why 2D simulation is discussed instead of "starting" from the simpler 1D case, i.e. for temporal disaggregation, which is important for several application problems. The Author should at least discuss the applicability to the framework to temporal disaggregation.

- It should be important also to mention one fundamental problem characterizing

MRC model, i.e. the stationarity of the disaggregated process. The Author mention the paper from Lombardo et al. (2017); yet in a previous paper (Lombardo et al., 2012) the authors demonstrated why MRC does not generate stationary processes. The Author should discuss the stationarity issue for the model proposed here and generally mention the possible limitation of the framework for temporal and spatial disaggregation.

- A figure could help to better explain the splitting rule.

- Line 17, page 5. How can we reproduce the stochastic properties of the process of alternation of wet and dry cells? If we do not explicitly model this process, how does the structure of this process depend on the model formulation?

- Lines 12-17, page 7. Is it possible to gain the same or similar advantage by using a different generator?

- Section 2.5. Since it is not clear which are the theoretical properties of the simulated random field (see general comment), it is difficult to follow this section.

- Lines 21-22. Is it possible to quantify this additional uncertainty?

- Lines 29-30. Based on this, it seems that the theoretical behavior of the simulated process is determined based on empirical reasons. Is this correct?

- Line 10, page 9. Smoother than the observed one, being based on interpolation . . . Thus, can this method be considered a disaggregation model? Instead of linear interpolation the Author could have considered kriging, which preserves the variogram.

- Lines 13-18, page 9. This configuration of the MRC is indeed not very common in the literature; how this configuration affects the reliability of the disaggregation model? Can we expect better results (as depicted in figures 7, 8 and 11) by using

the most common approach in the literature or one based on a larger number of parameters? Models can be compared also in case of a different complexity; furthermore, the parameters of disaggregation models could also be estimated by considering different additional properties of the generated process (e.g. the expected value of the number of wet cells in the spatial case or the dry spell average length in the case of time disaggregation).

- The calibration procedure is not totally clear to me; additional efforts are required to explain it in theory and practice.

- I'm not sure figure 6 is really useful. Are the numerical differences between EVA and traditional approach W values (as depicted in figure 6) really significant from a practical point of view?

- Given the large amount of observed rainfall fields, it could be of interest to understand how estimated model parameters depend on large scale event characteristics, so that a general parametrization valid for all the events can be found. This means calibrating the model based on observed small scale observations when available to apply the model to other events.

- Is it possible to compare simulated and observed fields in terms of the spatial structure of variability (spatial correlation) of the wet/dry alternation process? See also previous comment on this.

- The Author uses $R^2$ as a metric for model performance with respect to observation; why not using a different metric, not based on the normality assumption?

With appreciation, Elena Volpi

---

## Author Comment (AC1) · 15 Jan 2020

**Referee 1:**

In this paper the new approach to downscale precipitation field, based on equal-volume areas (EVAs) is evaluated. The topic of the paper is relevant for hydrology, meteorology and water management. The author compared the EVA method and "classical" discrete microcanonical random cascade and bilinear interpolation. The composition of the paper is valid. The methods are clearly described. The author mentioned both pros and cons of the EVA cascade generator and pointed the directions of further development. The results of EVA method application are worth to be published. However, some details should be corrected:

1. Page 6 line 30: "(. . .) from uniform (Olsson, 1998) to log-normal, Beta (. . .)". Please add reference to log-normal distribution usage and correct spelling to "beta".

Response: The spelling will be corrected and two additional references to Over and Gupta (1996) and Xu et al. (2015) will be added to the paper.

- Over, T. M. and Gupta, V. K. 1996: A space-time theory of mesoscale rainfall using random cascades, JGRA, vol.101, 2156-2202
https://agupubs.onlinelibrary.wiley.com/doi/abs/10.1029/96JD02033
- Xu, G. and Xu, X. and Liu, M. and Sun, Y. A. and Wang, K. 2015: Spatial Downscaling of TRMM Precipitation Product Using a Combined Multifractal and Regression Approach: Demonstration for South China, Water, vol.7(6), 3083-3102
https://www.mdpi.com/2073-4441/7/6/3083

2. Page 9 line 6: "While the EDA (. . .)" - Please check spelling.

Response: Thanks! The typo will be corrected during revision.

3. Page 10 line 11: "It shows large differences between estimated model parameters as a function of the methods and spatial resolutions." No relationship in tab. 2 is observed. Please paraphrase this sentence.

Response: The formulation will be changed to: "Retrieved model parameters are clearly sensitive to the spatial resolution of the input data, exhibiting different types of error patterns and biases as a function of the selected event and chosen cascade model."

4. Figure 4 - Please use the same bounds in vertical axis for the same parameter and do not use "true" adjective in horizontal axis title. Please name axis explicitly which parameter value is obtained for coarse-scale generator and which for radar data resolution.

Response: sure, no problem.

5. The author used plural form "we" multiple times, however, he is the only person sign to this paper.

Response: Sure, no problem. I will correct this during revision.

6. Figure 9 - Please correct the labels: "Sample gen" and "Best gen" – it is not known which is for 8x8 km resolution and which is for 1x1 km resolution.

Response: The labels will be adjusted accordingly.

7. Page 13 line 7: "Therefore, big local differences in scaling behavior exit within the field" - Please check spelling.

Response: Thanks! This was a typo.

8. Page 13 line 33: "Also, performance clearly decreases with intermittency" - Please add if the intermittency was calculated for empirical precipitation fields or for generated ones.

Response: The intermittency was obtained from the original high-resolution radar fields.

9. Page 14 line 23: "However, accuracy drops rapidly and large uncertainties are to be expected for such large downscaling ratios." - Please check grammar.

Response: I will replace the sentence by the following text: "However, the accuracy of downscaled rainfall fields for scale ratios of 256 or higher is likely to be low given that it is not always possible to reliably estimate the cascade generator from such coarse scale inputs."

10. Page 14 line 23: "(. . .) the EVA model is likely to be closer to the truth." - Please paraphrase this part of the sentence (the model results where compared with weather radar scans) - for instance "(. . .) the EVA model is likely to be closer to the observed precipitation fields".

Response: done.

---

## Author Comment (AC2) · 15 Jan 2020

**Referee 2:**

The manuscript proposes a new approach for rainfall disaggregation, based on the framework of Multiplicative Random Cascade (MRC) yet able to overcome its limitations, namely possible over-parametrization or limited accuracy in presence of intermittency (more specifically, for a large amount of zero values). The Author investigates the effectiveness of a disaggregation scheme based on the notions of scaling and Equal Volume Areas (EVA), with maximum three parameters that can be inferred directly from the coarse-scale data. Investigation is performed for 2D (spatial) rainfall fields. The proposed approach allows for intermittency simulation without explicitly coding zero values and their distribution and structure; on the other hand, the simulated rainfall field needs to be re-sampled in grid cells from variable to constant size (i.e., re-gridding).

**General comment:**

The manuscript is well written and organized, and the topic is surely of interest for the hydrologic community. Hence, I suggest to consider the manuscript for publication in HESS. Before this, more details are necessary to allow for reader understanding and repeatability of the work (that is also the application of the framework in different contests). The most important problem is that the theoretical properties of the simulated process, that are the marginal probability distribution function and the joint distribution, are not explicitly presented and discussed; these are fundamental to understand e.g. if the model is based on the same scaling property ruling the theoretical behavior of the traditional MRC. I have some additional concerns that are listed below. I hope they will be helpful for manuscript improvement.

Response: Thank you for taking the time to review this paper. I will add more details in Section 2.4 and 2.5 in order to better explain the mechanisms through which the EVA cascade generates intermittency and how model parameters can be estimated from the coarse-scale data. However, I don't think that it is possible (and that I would be capable) to derive meaningful, analytical expressions for characterizing the marginal probability distribution function of rainfall rates and dry/wet areas produced by the EVA cascade. Such theoretical derivations are beyond the scope of this paper and would depend a lot on the assumptions you make about the input field (e.g., distribution, stationarity, spatial structure, etc…). I understand the reviewer's interest for these issues but don't think that this is important for understanding the basic idea. So instead of making this very theoretical, I would like to stick to a more empirical approach and present the EVA cascade as a simple, alternative and convenient way to numerically transform a coarse-scale rainfall field into a high-resolution output. The application of this new cascade

model to a large number of rainfall fields shows promising performance, especially in the presence of intermittency.

**Specific and technical comments:**

• I'm curious about the reasons why 2D simulation is discussed instead of "starting" from the simpler 1D case, i.e. for temporal disaggregation, which is important for several application problems. The Author should at least discuss the applicability to the framework to temporal disaggregation.

Response: Thank you for this comment. Indeed, the same technique could be used to downscale time series based on the notion of inter-amount times (http://10.1175/JHM-D-15-0078.1). However, in this paper, I wanted to focus on designing a new spatial downscaling method, which is more challenging and interesting than time series downscaling because of the higher intermittency of 2D rainfall fields. This makes it easier to outline the advantages of using an adaptive sampling approach. But of course, the same approach could be used to downscale rainfall time series as well. Actually, in a previous paper (https://doi.org/10.1175/JHM-D-16-0221.1), I already outlined the superior scaling properties of inter-amount times in the presence of intermittency (without explicitly using them for downscaling). For completeness, I will add a few sentences about the downscaling of time series in the Discussion part. But this would have to be done within the context of another paper.

• It should be important also to mention one fundamental problem characterizing MRC model, i.e. the stationarity of the disaggregated process. The Author mentions the paper by Lombardo et al. (2017); yet in a previous paper (Lombardo et al., 2012) the authors demonstrated why MRC does not generate stationary processes. The Author should discuss the stationarity issue for the model proposed here and generally mention the possible limitation of the framework for temporal and spatial disaggregation.

Response: I will add a few lines about the stationarity issue in MRCs in the discussion part of the paper, together with a reference to Lombardo et al. 2012. But honestly, I don't see a lot of value in investigating stationarity issues within the context of this paper. The new EVA cascade proposed in this paper is just a convenient mathematical way to transform a coarse resolution input to a fine resolution output. Yes, there is a whole discipline in mathematics that deals with studying the theoretical properties of MRCs under idealized conditions. But rainfall is a complicated, highly intermittent and non-stationary process which means that these idealized conditions are very unlikely to be satisfied in reality. So instead of theorizing too much, I think there is more value in assessing the performance and practical advantages of the EVA cascade on real data, by comparing the EVA cascade with classical MRCs and discussing the pros and cons.

• A figure could help to better explain the splitting rule.

Response: Thanks for the suggestion. I will add a new figure during revision.

• Line 17, page 5. How can we reproduce the stochastic properties of the process of alternation of wet and dry cells? If we do not explicitly model this process, how does the structure of this process depend on the model formulation?

Response: The process by which intermittency is created during the cascade is explained at the end of Section 2.4, equations (9) and (10). The particularity of the EVA cascade is that some grid cells will get ``stuck'' during the downscaling, meaning that their area will converge to a fixed lower limit (different from zero) while the rainfall volume inside these grid cells will tend to zero. These are the areas which, after the final regridding, will correspond to dry areas. The spatial structure of the dry/wet areas and their frequency at a give scale is therefore controlled by the splitting rule or, equivalently, by the values of the three cascade generator parameters a, b and c. This was already explained in the text but may not have been formulated very clearly. I will add additional details during revision to make sure that this part is better explained, for example by splitting Section 2.4 in two parts and adding a new subsection for explaining how the cascade generator parameters influence the intermittency pattern.

• Lines 12-17, page 7. Is it possible to gain the same or similar advantage by using a different generator?

Response: Sure, many other similar distributions could be used with various degrees of interpretation for the parameters. I chose the logit-normal model because it was the easiest and most convenient that I could think of. I will add some text during the revision to mention some alternatives (e.g., the beta distribution).

• Section 2.5. Since it is not clear which are the theoretical properties of the simulated random field (see general comment), it is difficult to follow this section.

Response: Some additional details will be added to Section 2.4 during revision. This should make it easier to follow Section 2.5

• Lines 21-22. Is it possible to quantify this additional uncertainty?

Response: Yes, in theory, if we know the spatial autocorrelation structure of the rainfall field we could approximate the uncertainty due to linear interpolation. I will add some text to explain this. However, the quantification of this uncertainty and its incorporation into the estimation process goes beyond the scope of this paper.

• Lines 29-30. Based on this, it seems that the theoretical behavior of the simulated process is determined based on empirical reasons. Is this correct?

Response: I'm not sure to fully understand this comment. To clarify: the theoretical properties of the simulated rainfall fields are fully determined by the coarse-scale input data and the cascade generator model. The cascade generator model is fixed: it's a logit-normal distribution with mu=0 and variable standard deviation sigma. The parametric relation between sigma, the rainfall intensity R and area A in Equation 8 is based on empirical evidence from a large number of studies. It assumes that on average, spatial intermittency increases as we move to smaller scales and lower intensities. The parameters a, b and c needed to define the changes with A and R are inferred from the coarse scale data, depending on the properties of the considered rainfall field. The only empirical parts are therefore: 1) the choice of a logit-normal distribution for the cascade generator and 2) the assumption that the dependence of sigma on A and R can be approximated by Eq 8.

• Line 10, page 9. "Smoother than the observed one, being based on interpolation"
. . . Thus, can this method be considered a disaggregation model? Instead of linear interpolation the Author could have considered kriging, which preserves the variogram.

Response: Yes, the reviewer is right: bilinear interpolation with rescaling is not a disaggregation technique because it does not conserve the total rainfall amount in each coarse-scale grid cell. It only preserves the average intensity over the whole field, similarly to a canonical cascade. I will change the corresponding sentence during revision to clarify this point. Anyway, I don't think this is a big issue as bilinear interpolation is not the main focus of the paper and is only used as a first-order benchmark against which the improvements of the microcanonical cascades can be assessed. The suggestion to use kriging is interesting. I had thought of this initially and indeed, there are some situations in which kriging works better than bilinear interpolation. But there wasn't a big improvement in performance and any simple micro-canonical cascade would do better than kriging. In addition, there are many downsides to using kriging in downscaling applications. The first is that kriging is a linear estimator whereas bilinear interpolation is not. The second is that kriging is slower than bilinear interpolation. The third is that kriging makes pretty strong assumptions about the data. In particular, the mean and the variance of the rainfall process are not allowed to change with the location. This is often not true in practice, especially in the presence of intermittency. For more information about this, please have a look at my 2014 paper in JHM: "Non-stationarity in intermittent rainfall: the dry drift" (http://10.1175/JHM-D-13-095.1). Lastly, it must be said that kriging requires the estimation and fitting of a variogram model from the data. This creates all sorts of numerical issues and

complications, with many exceptions and special cases. If the spatial structure is complicated and the variogram model do not fit the sample variogram very well, chances are that kriging will not be as good as anticipated. By contrast, bilinear interpolation is a non-parametric and local method that makes no assumptions about the data and is therefore hard to beat in practice. For all these reasons, I think it is better to stick with bilinear interpolation as the benchmark rather than kriging.

• Lines 13-18, page 9. This configuration of the MRC is indeed not very common in the literature; how does this configuration affect the reliability of the disaggregation model Can we expect better results (as depicted in figures 7, 8 and 11) by using the most common approach in the literature or one based on a larger number of parameters. Models can be compared also in case of a different complexity; furthermore, the parameters of disaggregation models could also be estimated by considering different additional properties of the generated process (e.g. the expected value of the number of wet cells in the spatial case or the dry spell average length in the case of time disaggregation).

Response: Sure, one could easily improve the performance of the classical MRC by introducing additional parameters and performing a separation between dry and wet components. There is little doubt that such a state-of-the-art model with 6-7 parameters would outperform the simple EVA cascade proposed in this paper. At the same time, I don't think that such comparisons are really fair and helpful at this stage. Future model developments directed at improving the EVA cascade with the help of additional parameters and more flexible generator models should be part of a follow-up study. Performance is obviously important. But the primary goal of this paper is to introduce a new conceptual way of approaching the downscaling problem in the presence of intermittency. The comparisons with the MRC are done to demonstrate potential and gain insight into the pros and cons of each approach. Optimization is not the primary concern here and can always be done later.

• The calibration procedure is not totally clear to me; additional efforts are required to explain it in theory and practice.

Response: I will add more details about this during the revision. The model parameters are estimated in a very natural way, by aggregating the coarse-scale field to larger scales and studying the splitting behavior of larger grid cells.

• I'm not sure figure 6 is really useful. Are the numerical differences between EVA and traditional approach W values (as depicted in figure 6) really significant from a practical point of view?

Response: Yes, of course! They are the main reason why, for similar grid cell areas and intensities, the EVA model tends to have a generator with lower variance. This is crucial for understanding why the EVA method produces slightly smoother fields and lower, more realistic rainfall extremes.

• Given the large amount of observed rainfall fields, it could be of interest to understand how estimated model parameters depend on large scale event characteristics, so that a general parametrization valid for all the events can be found. This means calibrating the model based on observed small scale observations when available to apply the model to other events.

Response: Thank you for the suggestion. I have already looked at this but did not find any strong link between large-scale properties such as intermittency, variance or decorrelation range and the parameter values of the generator. Also, this will depend a lot on which sensor you used for measuring the rainfall field and would not be easily transferable to other regions or datasets. One promising idea though could be to model the bias affecting the estimated model parameters and compensate for it during the parameterization step. This can significantly improve the performance for larger downscaling ratios. I will add a short paragraph about this in the discussion.

• Is it possible to compare simulated and observed fields in terms of the spatial structure of variability (spatial correlation) of the wet/dry alternation process? See also previous comment on this.

Response: Yes, sure! I will add a figure similar to Fig10 but for the dry/wet transitions together with some explanations.

• The Author uses R2 as a metric for model performance with respect to observation; why not using a different metric, not based on the normality assumption?

Response: Actually, the coefficient of determination (R2) does not require any normality assumption. The normality assumption is only needed for the interpretation of the R2 within the framework of a linear regression model (e.g., in terms of explained variance). But this is not the goal here and I just use the R2 as an easy way to measure the relative goodness of fit and support the visual assessment of the scatterplots. This is not critical and any other related metric could be used for this without affecting the conclusions.

---

## Author Response (AR1)

**Referee 1:**

In this paper the new approach to downscale precipitation field, based on equal-volume areas (EVAs) is evaluated. The topic of the paper is relevant for hydrology, meteorology and water management. The author compared the EVA method and "classical" discrete microcanonical random cascade and bilinear interpolation. The composition of the paper is valid. The methods are clearly described. The author mentioned both pros and cons of the EVA cascade generator and pointed the directions of further development. The results of EVA method application are worth to be published. However, some details should be corrected:

1. Page 6 line 30: "(. . .) from uniform (Olsson, 1998) to log-normal, Beta (. . .)". Please add reference to log-normal distribution usage and correct spelling to "beta".

> Response: The spelling has been corrected and two additional references to Over and Gupta (1996) and Xu et al. (2015) have been added to the paper.
>
> - Over, T. M. and Gupta, V. K. 1996: A space-time theory of mesoscale rainfall using random cascades, JGRA, vol.101, 2156-2202
> https://agupubs.onlinelibrary.wiley.com/doi/abs/10.1029/96JD02033
> - Xu, G. and Xu, X. and Liu, M. and Sun, Y. A. and Wang, K. 2015: Spatial Downscaling of TRMM Precipitation Product Using a Combined Multifractal and Regression Approach: Demonstration for South China, Water, vol.7(6), 3083-3102
> https://www.mdpi.com/2073-4441/7/6/3083

2. Page 9 line 6: "While the EDA (. . .)" - Please check spelling.

> Response: Thanks! The typo has been corrected during revision.

3. Page 10 line 11: "It shows large differences between estimated model parameters as a function of the methods and spatial resolutions." No relationship in tab. 2 is observed. Please paraphrase this sentence.

> Response: The formulation has been changed to:
>
> *"Retrieved model parameters are clearly sensitive to the spatial resolution of the input data, exhibiting different types of error patterns and biases as a function of the selected event and chosen cascade model."*

4. Figure 4 - Please use the same bounds in vertical axis for the same parameter and do not use "true" adjective in horizontal axis title. Please name axis explicitly which parameter value is obtained for coarse-scale generator and which for radar data resolution.

       Response: The plots now have the same axes for each cascade/parameter and the axis titles have been changed.

5. The author used plural form "we" multiple times, however, he is the only person sign to this paper.

       Response: This has been corrected during revision.

6. Figure 9 - Please correct the labels: "Sample gen" and "Best gen" – it is not known which is for 8x8 km resolution and which is for 1x1 km resolution.

       Response: The labels were changed.

7. Page 13 line 7: "Therefore, big local differences in scaling behavior exit within the field" - Please check spelling.

       Response: Thanks! This was a typo.

8. Page 13 line 33: "Also, performance clearly decreases with intermittency" Please add if the intermittency was calculated for empirical precipitation fields or for generated ones.

       Response: The intermittency was obtained from the original high-resolution radar fields at 1x1 km. This is not clearly mentioned in the text.

9. Page 14 line 23: "However, accuracy drops rapidly and large uncertainties are to be expected for such large downscaling ratios." - Please check grammar.

       Response: The sentence has been reformulated:

       *"However, the accuracy of downscaled rainfall fields for scale ratios of 256 or higher is likely to be low given that it is not always possible to reliably estimate the cascade generator from such coarse scale inputs."*

10. Page 14 line 23: "(. . .) the EVA model is likely to be closer to the truth." Please paraphrase this part of the sentence (the model results where compared with weather

radar scans) - for instance "(. . .) the EVA model is likely to be closer to the observed precipitation fields".

Response: Done!

**Referee 2:**

**General comment:**

The manuscript is well written and organized, and the topic is surely of interest for the hydrologic community. Hence, I suggest to consider the manuscript for publication in HESS. Before this, more details are necessary to allow for reader understanding and repeatability of the work (that is also the application of the framework in different contests). The most important problem is that the theoretical properties of the simulated process, that are the marginal probability distribution function and the joint distribution, are not explicitly presented and discussed; these are fundamental to understand e.g. if the model is based on the same scaling property ruling the theoretical behavior of the traditional MRC. I have some additional concerns that are listed below. I hope they will be helpful for manuscript improvement.

> Response: Thank you for reviewing this paper. I have taken into account all your comments and revised the paper accordingly. The major changes are:
>
> - More details in sections 2.4, 2.5, 2.6 to explain how the EVA cascade works, how it generates intermittency and how to estimate model parameters.
> - 4 new figures for illustrating the splitting rule (Fig.2), the convergence of the cascade (Fig.4), estimation of empirical breakdown coefficients (Fig.5) and spatial structure of occurrence field (Fig 14).
> - Additional remarks in the Discussion/Conclusion sections (see below)
>
> Comments that have not been considered:
> - Analytical expressions for characterizing the marginal probability distribution function of rainfall rates and dry/wet areas produced by the EVA cascade. Such theoretical derivations are beyond the scope of this paper and would depend a lot on the initial assumptions about the input field (e.g., distribution, stationarity, spatial structure, etc…). I understand the reviewer's interest for these issues but don't think that this is crucial for understanding the basic idea.

**Specific/technical comments:**

• I'm curious about the reasons why 2D simulation is discussed instead of "starting" from the simpler 1D case, i.e. for temporal disaggregation, which is important for several application problems. The Author should at least discuss the applicability to the framework to temporal disaggregation.

Response: Indeed, the same technique could be used to downscale time series based on the notion of inter-amount times (http://10.1175/JHM-D-15-0078.1). This is now clearly mentioned in the Discussion section.

*"Similarly to classical multiplicative random cascades, the EVA cascade can be applied to downscale time series, spatial and space-time data. For time series, the equivalent formalism is given by the notion of ``equal-volume times'', also known as inter-amount times (Schleiss and Smith 2016; Schleiss, 2017). Future work will therefore be directed at exploiting the superior scaling properties of inter-amount times to downscale time series of intermittent rainfall as well as combining IATs with EVAs to design more general downscaling schemes for space-time data."*

However, in this paper, I wanted to focus on spatial downscaling, which is more challenging than time series because of the higher intermittency of 2D fields.

• It should be important also to mention one fundamental problem characterizing MRC model, i.e. the stationarity of the disaggregated process. The Author mentions the paper by Lombardo et al. (2017); yet in a previous paper (Lombardo et al., 2012) the authors demonstrated why MRC does not generate stationary processes. The Author should discuss the stationarity issue for the model proposed here and generally mention the possible limitation of the framework for temporal and spatial disaggregation.

Response: I added a few lines about the stationarity issue in MRCs in the discussion part of the paper, together with a reference to Lombardo et al. 2012.

*"The third issue that needs to be mentioned relates to the assumption that the cascade generator model is stationary and, in particular, location invariant (i.e., that the same splitting rules apply to all pixels, independently of their location). This may not necessarily be valid for highly heterogeneous fields, as highlighted by the poor performance and inconsistent behavior of the cascade models during event 3. The key point here is that there might be specific areas within a rainfall field where the scaling properties are different from the rest (e.g., stratiform vs convective areas). Similarly, the scaling properties and spatial variability within individual rainfall cells might be very different from the average variability observed over a large collection of rain cells. Also, elements belonging to larger-scale structures might evolve together in a more coherent and predictable way than expected based on their size and intensity. One possible solution to overcome this problem would be to define multiple local generators instead of a single universal one. But this is a very challenging problem that requires more research, including the ability to automatically detect strong local variations in scaling properties to help pinpoint problematic regions and come up with a better approach. Also, the use of multiple generators would require additional model parameters, which is not necessarily desirable and should only be considered when absolutely necessary (e.g., for example to account for strong orographic effects). On a more theoretical level, one should also point out that even if the cascade generator is perfectly stationary, the final disaggregated fields (or time series) obtained after applying the cascade are likely to be non-stationary with location and time-dependent autocorrelation structures (Lombardo, 2012)"*

Stationarity obviously plays an important theoretical role in random cascades. But I don't see a lot of value in investigating these theoretical issues in more depth within the context of this paper. Instead, I prefer to focus on assessing the performance and merits/limitations of the EVA and classical MRCs on real data.

• A figure could help to better explain the splitting rule.

Response: Thank you for the suggestion. A new figure (Fig 2) has been added to better illustrate the splitting rule, together with a short explanation in the text.

*"The splitting rule is a set of instructions for determining which side gets assigned the lowest area or, equivalently, the highest rainfall intensity. To preserve the overall spatial structure and coherency of the rainfall field during downscaling, knowledge about the rainfall intensity in surrounding grid cells is required. This is achieved by performing inverse distance interpolation of the coarse-scale rainfall intensity field on the left/right (for horizontal splits) or top/bottom (for vertical splits) sides of each grid cell. At each split, the side with the highest interpolated rainfall value gets assigned the largest intensity (i.e., the smallest area). An example of this principle is shown in Figure 2 for a single grid cell (in bold at the center of the figure) with area A surrounded by 7 grid cells with different areas and intensities.*

• Line 17, page 5. How can we reproduce the stochastic properties of the process of alternation of wet and dry cells? If we do not explicitly model this process, how does the structure of this process depend on the model formulation?

Response: This was already explained in the text but may not have been formulated very clearly. I have added a new figure (Fig 4) to illustrate the process and some clarifications in the text (Sections 2.4 and 2.5).

*"[…] The final result of this process is a bounded cascade in which some grid cells have areas converging to a fixed value (or equivalently, intensity converging to zero) while all other grid cells have rainfall rates converging to a strictly positive value. Figure 4 illustrates this process, showing how the area of some small grid cells gets stuck' during the cascade while all the others end up splitting more and more evenly. However, note that since the weights are drawn at random, the process only convergences in a probabilistic sense, that is, on average, over a large number of cascade levels and splits. The condition b>c in Equation (8) is used to ensure convergence by preventing any uncontrolled increases in rainfall intensities from one level to another of the cascade. Indeed, the generator is built in such a way that whenever the intensity in a grid cell increases, the standard deviation of the generator decreases. This forces subsequent splits to be more even and reduces the probability of seeing any further increases in intensity at the next levels. This also means that the largest changes in rainfall intensities tend to occur at the earlier stages of the cascade, when the variance of the generator is still large. The magnitude of the random fluctuations then progressively decreases (at a rate that depends on the values of a, b and c), and intensities quickly converge to a fixed value. This can be seen as a strength, as it means that the cascade is very stable and can be stopped after a small number of iterations (i.e., as soon as the output has stabilized). However, it can also be a*

*disadvantage, as fast convergence means that the EVA cascade is more likely to underestimate small-scale variability (especially for large downscaling ratios).*

• Lines 12-17, page 7. Is it possible to gain the same or similar advantage by using a different generator?

> Response: Sure, many other similar distributions could be used with various degrees of interpretation for the parameters. This is now clearly mentioned in the discussion part:
>
> *"Also, different cascade distribution models could be used with various degrees of interpretation for the parameters. In this work, the logit-normal model was chosen because it was the easiest and most convenient while providing a reasonable fit to empirical cascade weights. However, other more flexible distribution models could be used (e.g., the beta distribution)"*

• Section 2.5. Since it is not clear which are the theoretical properties of the simulated random field (see general comment), it is difficult to follow this section.

> Response: I've done my best to clarify the way the EVA cascade works in Sections 2.3, 2.4 and 2.5. Also, the new figures added during revision should help. However, as I explained in response to the general comment, I don't think that further theoretical derivations are necessary to understand the general idea.

• Lines 21-22. Is it possible to quantify this additional uncertainty?

> Response: Yes, in theory, if we know the spatial autocorrelation structure of the rainfall field we could approximate the uncertainty due to linear interpolation. I've added a few lines in the text to explain this.
>
> *The main drawback compared with the classical approach is that due to the fixed grid spacing, the sub-areas can not be determined exactly but must be approximated by linear interpolation, similarly to the procedure described in Equation (4) of Schleiss (2017). Figure 5 shows an example of this for a single grid cell of size 8x8 for both horizontal and vertical splits. For the vertical split, the two sub-grid cells are of size 4.32x8 and 3.68x8. The first dimension (i.e., 4.32) is obtained by interpolating the rainfall amount contained in the smaller grid cell of size 4x8 (containing slightly less than half the amount) and the one immediately above of size 5x8 (which contains more than half). The additional interpolation step means that the empirical breakdown coefficients of small grid cells will be affected by larger sampling uncertainties compared with large grid cells. In theory, one could calculate the local spatial autocorrelation structure of the rainfall field to estimate the uncertainty due to linear interpolation. However, the quantification of this uncertainty and its incorporation into the estimation process goes beyond the scope of this paper and will be ignored here.*

• Lines 29-30. Based on this, it seems that the theoretical behavior of the simulated process is determined based on empirical reasons. Is this correct?

> Response: Partly, yes. The theoretical properties of the simulated rainfall fields are fully determined by the coarse-scale input data and the cascade generator model. The most empirical aspect is the relation linking sigma to the rainfall intensity R and area A in Equation (8). The latter was inferred from a large number of radar snapshots and is based on the (reasonable) assumption that on average, spatial variability within a grid cell increases with area and decreases with intensity. The parameters a, b and c which define the change of sigma with A and R are inferred from the coarse scale data and tuned to the properties of individual rainfall fields.

• Line 10, page 9. "Smoother than the observed one, being based on interpolation" . . . Thus, can this method be considered a disaggregation model? Instead of linear interpolation the Author could have considered kriging, which preserves the variogram.

> Response: Yes, the reviewer is right: bilinear interpolation with rescaling is not a disaggregation technique because it does not conserve the total rainfall amount in each coarse-scale grid cell. I've revised the text to highlight this point:

> *"While the EVA downscaling technique is the main focus of this paper, two additional spatial downscaling techniques were considered for comparison purposes. The first is bilinear interpolation, implemented in the function ``interp.surface()'' of the R package ``fields''. Bilinear interpolation is a deterministic non-parametric downscaling method. It makes no assumption about the structure and distribution of the data, making it very robust. However, because it is an interpolation technique, it tends to generate fields that are too smooth compared with the observations. Note that strictly speaking, bilinear interpolation is not a disaggregation technique because it does not conserve the total rainfall amount in each coarse-scale grid cell. However, the interpolated values can always be rescaled such that that the average rainfall intensity over the whole domain is preserved, similarly to a canonical cascade. This technicality is not crucial here since bilinear interpolation is not the main focus of the paper and is only used as a rough baseline against which the added-value of the random cascade models can be assessed."*

> The suggestion to use kriging is interesting. I had thought of this initially and indeed, there are some situations in which kriging works better than bilinear interpolation. But overall, there wasn't a clear improvement in performance. I've added a sentence in the text to mention this:

*"Also note that other interpolation techniques such as kriging were explored. But the downscaled fields were still too smooth and no clear improvement in performance was observed compared with bilinear interpolation."*

The main reason why kriging is not necessarily better is that it makes pretty strong assumptions about the data (i.e., essentially, the mean and the variance of the rainfall process are not allowed to change with location). This is often too restrictive in practice, especially in the presence of intermittency. For more information about this issue, please have a look at my 2014 paper in JHM: "Non-stationarity in intermittent rainfall: the dry drift" (http://10.1175/JHM-D-13-095.1). By contrast, bilinear interpolation is a non-parametric and local method that makes no assumptions about the data and is therefore hard to beat in practice.

• Lines 13-18, page 9. This configuration of the MRC is indeed not very common in the literature; how does this configuration affect the reliability of the disaggregation model Can we expect better results (as depicted in figures 7, 8 and 11) by using the most common approach in the literature or one based on a larger number of parameters. Models can be compared also in case of a different complexity; furthermore, the parameters of disaggregation models could also be estimated by considering different additional properties of the generated process (e.g. the expected value of the number of wet cells in the spatial case or the dry spell average length in the case of time disaggregation).

Response: Sure, one could easily improve the performance of the classical MRC by introducing additional parameters and performing a separation between dry and wet components. There is little doubt that such a state-of-the-art model with 6-7 parameters would outperform the simple EVA cascade proposed in this paper. This is now clearly stated in the conclusions. At the same time, I don't think that such comparisons are really fair and helpful at this stage. Future model developments directed at improving the EVA cascade with the help of additional parameters and more flexible generator models should be part of a follow-up study. Performance is obviously important. But the primary goal of this paper is to introduce a new conceptual way of approaching the downscaling problem in the presence of intermittency. Optimization is not the primary concern here and can always be done later.

*"Obviously, one can always improve the performance by introducing more model parameters or tuning them to individual cases. Similarly, one could easily increase the performance of the classical cascade by performing a separation between dry and wet components before disaggregation. There is little doubt that such a state-of-the-art model with 6-7 parameters would outperform the simple EVA cascade proposed in this paper. At the same time, such comparisons are not really fair and helpful at this stage, as optimization was not the primary objective of this paper and the EVA model should not be seen as a competitor designed to*

*replace traditional cascades but rather as a new complementary tool for modelers to deal with intermittency and get new insight into the complex spatio-temporal organization of rainfall across scales."*

• The calibration procedure is not totally clear to me; additional efforts are required to explain it in theory and practice.

> Response: I've added a new figure (Fig 5) to illustrate this, together with some explanations in Section 2.6

• I'm not sure figure 6 is really useful. Are the numerical differences between EVA and traditional approach W values (as depicted in figure 6) really significant from a practical point of view?

> Response: Yes, of course! They are the main reason why, for similar grid cell areas and intensities, the EVA model tends to have a generator with lower variance. This is crucial for understanding why the EVA method produces slightly smoother fields and lower, more realistic rainfall extremes.

• Given the large amount of observed rainfall fields, it could be of interest to understand how estimated model parameters depend on large scale event characteristics, so that a general parametrization valid for all the events can be found. This means calibrating the model based on observed small scale observations when available to apply the model to other events.

> Response: Thank you for the suggestion. I have already looked at this but did not find any strong link between large-scale properties such as intermittency, variance or decorrelation range and the parameter values of the generator. Also, this will depend a lot on the characteristics of the sensor used for measuring the rainfall field and would not be easily transferable to other regions or datasets. One promising idea though could be to model the bias affecting the estimated model parameters and compensate for it during the parameterization step. This can significantly improve the performance for larger downscaling ratios. However, more research is needed to figure out the best way to do this.

> *"Another possibility would be to design flexible climatological generator values that can be adjusted depending on rainfall type and large-scale properties (e.g., intensity, intermittency, range), an approach that may be more flexible while limiting sampling issues. Preliminary work performed within this study (not shown) suggests that this may be promising for larger downscaling ratios as cascade parameters often tend to be correlated with each other or to large-scale rainfall properties."*

• Is it possible to compare simulated and observed fields in terms of the spatial

structure of variability (spatial correlation) of the wet/dry alternation process? See also previous comment on this.

> Response: Yes, sure. An additional figure (Fig 14) has been added to show the spatial semivariance of the downscaled occurrence field (for events 1-4). It shows that the EVA cascade tends to produce occurrence fields with more realistic structures (on average) but also highlights that there are events for which the classical cascade is better at reproducing intermittency.

• The Author uses R2 as a metric for model performance with respect to observation; why not using a different metric, not based on the normality assumption?

> Response: Actually, the coefficient of determination (R2) does not require any normality assumption. The normality assumption is only needed for the interpretation of the R2 within the framework of a linear regression model (e.g., in terms of explained variance). But this is not the goal here and I just use the R2 as an easy way to measure the relative goodness of fit and support the visual assessment of the scatterplots. This is not critical and any other related metric could be used for this without affecting the conclusions.

[revised manuscript text omitted]

---

## Referee Report (RR1)

**"A new discrete multiplicative random cascade model for downscaling intermittent rainfall fields"**
**by Marc Schleiss - Referee Report, 2 May 2020**

In this paper the new approach to downscale precipitation field, based on equal-volume areas (EVAs) is evaluated.

The subject of the research – spatial modelling of precipitation fields - is a relevant issue for hydrology, meteorology and water management. The author faced with downscaling intermittent rainfall fields, which is nontrivial problem. EVA model is shown as a new complementary tool for modellers to deal with intermittency.The author compered EVA method and "classical" discrete multi-canonical random cascade and described the advantages and weaknesses of both. Moreover, he pointed out the complementarity of this two approaches. His research might open new opportunities for quality control and bias corrections of downscaled fields.

The composition of the paper is valid. The methods are clearly described. The results of EVA method application are worth to be published.